# Benchmarking Optimizers for Large Language Model Pretraining

## Abstract

The recent development of Large Language Models (LLMs) has been accompanied by an effervescence of novel ideas and methods to better optimize the loss of deep learning models. Claims from those methods are myriad: from faster convergence to removing reliance on certain hyperparameters. However, the diverse experimental protocols used to validate these claims make direct comparisons between methods challenging. This study presents a comprehensive evaluation of recent optimization techniques across standardized LLM pre-training scenarios, systematically varying model size, batch size, and training duration. Through careful tuning of each method, we provide guidance to practitioners on which optimizer is best suited for each scenario. For researchers, our work highlights promising directions for future optimization research. Finally, by releasing our code and making all experiments fully reproducible, we hope our efforts can help the development and rigorous benchmarking of future methods.

## 1  Introduction

Over the past five years, Large Language Models (LLMs) [15, 59, 22, 48] have shown growth in performance and size, demonstrating proficiency in various downstream tasks [80, 7, 85]. The success of LLM pretraining hinges on three key pillars: high-quality data [65, 44], architectural innovations [31, 15], and scalable optimization techniques.

Among these, the choice of optimizer has remained notably consistent in recent years, with `Adam(W)` [38, 50] dominating deep learning for nearly a decade. However, recent advances [33, 47, 84, 62, 66, 17] challenge this status quo, offering alternatives that surpass `AdamW` in speed, communication efficiency [1] or final downstream performance on various benchmarks [12, 37], particularly for autoregressive language modeling [70]. Despite these innovations, current benchmarks and ablation studies [96, 34] remain narrow in scope, often examining only isolated aspects of optimizer design. This lack of systematic comparison makes it difficult to obtain trustworthy insights for practitioners, or identify the next promising research directions.

In this work, our goal to revisit the problem of benchmarking optimizers for LLM pretraining. We do so through standardized experiments which vary important parameters such as batch size, model size, and the number of training iterations. This allows us to formulate an up-to-date list of best-performing methods for the community of researchers and practitioners. We demonstrate the efficiency of each considered method through careful tuning, and present insightful ablations along the way. Furthermore, we provide a set of best practices for LLM pretraining that are applicable regardless of the optimizer chosen.

We summarize our contributions as follows:

**(Contribution 1)** We conduct the first large-scale, controlled benchmark of 11 different optimization methods across diverse LLM training scenarios. A fair comparison is ensured by precise accounting for compute costs, and extensive hyperparameter tuning. We identify optimal optimizer choices in several relevant training regimes, for both dense and MoE architectures.

**(Contribution 2)** We perform comprehensive ablations of critical training hyperparameters— including warmup duration, initialization schemes, gradient clipping, final learning rates, and learning rate scheduler choices—providing actionable insights for optimizing LLM training in practice.

**(Contribution 3)** We open-source our full benchmarking toolkit, including training scripts, evaluation pipelines, and hyperparameter configurations, to enable reproducible research and facilitate future optimizer development.

**For practitioners**, our work provides an evidence-based answer to the burning question: "*Is Adam still the most effective optimizer in the age of LLMs, or can we achieve better performance at scale with novel optimizers?*". **For researchers**, our work delivers a unified benchmarking framework for LLM pretraining, along with extensive ablation studies which systematically evaluate both popular and overlooked optimizer designs—revealing previously unexplored tradeoffs between efficiency, stability, and final model performance. Overall, our findings not only challenge long-held assumptions about optimizer selection but also establish a foundation for future advances in large-scale model training. By bridging the gap between theoretical innovation and practical deployment, this work aims to accelerate progress in both research and industry applications of LLM training.

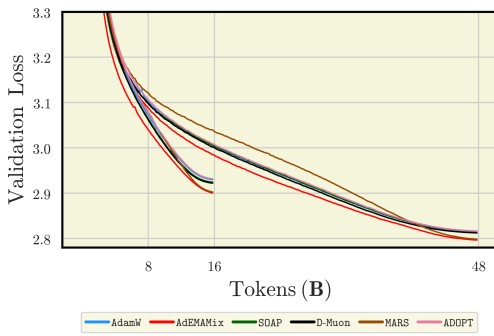

Figure 1: **A comparison of leading optimizers, for training a 720M parameter LLM.**

# 2   Background & Related Work

**Optimizers.**   While computer vision models often show comparable performance between SGD [72] and AdamW [94], the landscape differs dramatically in LLM training. Recent work [95] demonstrates that adaptive methods like AdamW provide substantially better optimization characteristics for transformer-based language models. The question of why AdamW works so well has been a long-standing topic of research [2, 60, 93, 43, 41]. Modern methods often inherit AdamW's core ideas in their structure, such as ADOPT [83] and AdEMAMix [62]. ADOPT has been motivated by solving long-standing convergence issues in AdamW. By normalizing the second-order moment prior to the momentum update, they eliminate the non-convergence issues of AdamW on smooth non-convex functions. Meanwhile AdEMAMix extends AdamW with an additional slower momentum buffer, i.e. a slower exponential moving average (EMA), which allows the use of much larger momentum values, accelerating convergence.

One interpretation of AdamW's effectiveness lies in its sign-based update [42]: without the exponential moving average (EMA), AdamW resembles signSGD [6]. Recent work [96, 36] has shown that Signum (signSGD with momentum), can perform comparably to AdamW. Earlier, the community also discussed Lion [9], a method with a similar sign-based structure. Signum and Lion offer memory benefits due to the use of only a single instead of Adam's two buffers for optimizer states.

Another family of methods stems from AdamW's approximate second-order structure, where the diagonal of the Fisher information matrix or other preconditioning approaches [52, 24] are used as the second moment estimate. This idea has given rise to Sophia [46], SOAP [84], and, to some extent, Muon [33].

The parameter-free concept [61] has led to the development of Schedule-Free AdamW (SF-AdamW) [17] and Prodigy [54]. These optimizers do not require a decreasing learning rate schedule, making them relevant for continual training. Last but not least, MARS [88], builds upon this line of research and incorporates a variance reduction mechanism in its update rule.

**Benchmarks.** To a large extent, the benchmarking setup determines the final conclusions. Some benchmarks are designed for short speedruns in terms of training or validation loss [32], while others focus on a downstream target metric after training [96, 12, 76]. Methods that perform well in short speedruns might not be optimal for longer training horizons as in real LLM training runs (see Figure 3). "*But what constitutes a sufficiently long horizon?*" "*What should be the compute budget for LLM training?*" These are questions explored by scaling laws [35]. Early benchmarks for optimizers and other ablation studies often rely on Chinchilla scaling laws [26] with a ratio of roughly 20 tokens per parameter (TPR) needed for pretraining. However, recent research [69, 74] argues that this is far from sufficient for production-ready models.

Another important issue is the choice of loss function. Recent setups have been using an auxiliary z-loss [86, 11] in addition to cross-entropy, which requires further investigation. We believe this choice is influenced by the use of the OLMo [58] codebase, which we also address in our work.

Additionally, we found that previous setups for comparing optimizers do not align with recent best practices regarding weight decay, learning rate decay, and overall hyperparameter tuning. All of these questions are revisited in our work.

## 3 Experimental Setup

**Notations.** We use the following notations. Let $\gamma$ be the learning rate, $\lambda$ the weight decay coefficient, and $T$ the total number of iterations. Momentum-related parameters are represented by the symbol $\beta$.

**Models & Data.** For most experiments, we use a Llama-like transformer [48] architecture, including SwiGLU activations [77], RMSNorm [91], and RoPE embeddings [81]. We experiment with four sizes of models: 124M, 210M, 583M, 720M. We train on a $100B$ tokens[1] subset of FineWeb [64]. It consists of a cleaned and deduplicated corpus for LLM pretraining, which we tokenize using the GPT-2 tokenizer prior to splitting into train and validation sequences. MoE setup described in

**Iterations & Batch size.** Throughout our experiments, we use a sequence length of 512 tokens. For clarity, we often report the batch size in tokens by writing *Batch size×sequence length*. For the 124M model, we use batch sizes of $32 \times 512 = 16$k, $256 \times 512 = 131$k, and $512 \times 512 = 262$k tokens; for the 210M model, we use a batch size of $256 \times 512 = 131$k; and for 583M model, we leverage the batch sizes of $1024 \times 512 = 524$k and $3936 \times 512 = 2$M tokens. Depending on the model size, we vary the number of iterations — also measured in tokens for compatibility with scaling laws and to accommodate different batch size settings. We train 124M and 210M models for equal durations of $\{1, 2.1, 4.2, 6.3, 8.4, 16.8\}$B tokens. This corresponds to $T \in \{64, 128, 256, 384, 512, 1024\}$k iterations for a batch size of 32, and $T \in \{8, 16, 32, 48, 64, 128\}$k iterations for a batch size of 256. For 583M models, we train on $\{13, 32\}$B tokens, corresponding to $T \in \{6.5, 16\}$k iterations, resp. $T \in \{25, 61.5\}$k iterations, for a batch size of 3936, resp. 1024. In the setup with 720M model, we have $T \in \{8, 16, 48\}$k iterations for a batch size of 1M tokens. Thus, for all model scales, we include both Chinchilla optimal lengths of training and beyond. More details are available in Appendix C.

**Loss.** We train using the classical cross-entropy next token prediction loss. Some prior works introducing optimizers use a z-loss in addition to cross-entropy [30, 11, 86, 84, 96]. We found that this has little impact and, therefore, do not use z-loss. An ablation showing results with and without z-loss is in the appendix.

**Hyperparameter Tuning.** Training LLMs is a computationally intensive task. As a guidance, practitioners often rely on insights gathered at lower scales, scaling laws, and other rules [87, 18]. It is also commonplace to run experiments for only a shorter duration of training, as a way to test certain hyperparameters prior to extending the training horizon to more iterations. Because a full grid search over every hyperparameter, for each setting and optimizer, would be too costly, we resort to a similar approach. More precisely, for each model size, batch size, and optimizer, we tune optimization hyperparameters extensively for a number of training tokens which is near-Chinchilla optimal. We then keep those hyperparameters when we increase the number of iterations. While we found that the sensitivity to several hyperparameters can change as we increase the training horizon, we found this approach simple and yet effective. The hyperparameters being considered depend on the optimizer. We proceeded from small to large model scale, and used insights gathered at smaller scales

---

[1] https://huggingface.co/datasets/HuggingFaceFW/fineweb

to guide the hyperparameter search at larger scales. Our hyperparameter sweeps are summarized in Appendix D. We present the clarifications regarding the connection between the number of iterations and tokens for different batch size settings as well as the Chinchilla optimal length of training for our models in Tables 3 and 5. As learning rate schedulers, we compare cosine [49], linear and warmup-stable-decay (WSD) [27, 90, 28]. Unless specified, we use a cosine scheduler. Results with WSD and linear schedulers are discussed in Section 4. Recent works also emphasize the importance of sufficiently decaying the learning rate [4, 75, 28]. As such, we take care to decay to $0.01 \times \gamma$ instead of the often used $0.1 \times \gamma$. To give an idea of how much effort was put into tuning each method, across all model sizes, batches and iterations, we trained a total of 2400 models, and have spent roughly 30000 GPU hours.

**Optimizers.** Here is a list of the optimizers we considered in our work. For each algorithm, we write in parentheses the optimizer-specific hyperparameters we tuned: $\texttt{AdamW}(\beta_1, \beta_2)$, $\texttt{SOAP}(\beta_1, \beta_2)$ and preconditioning frequency, $\texttt{Lion}(\beta_1, \beta_2)$, $\texttt{MARS}(\eta, \beta_1, \beta_2)$ and Newton-Schulz hyperparameters, $\texttt{ADOPT}(\beta_1, \beta_2)$, $\texttt{Signum}(\beta)$, $\texttt{Prodigy}(\beta_1, \beta_2)$, $\texttt{SF-AdamW}(\beta_1, \beta_2)$, $\texttt{Muon}(\gamma^\texttt{M}, \beta, \beta_1, \beta_2)$, $\texttt{Sophia}(\rho, \beta_1, \beta_2)$, $\texttt{AdEMAMix}(\beta_1, \beta_2, \beta_3, \alpha)$. When an optimizer has several momentum variants e.g. Nesterov [57] or Polyak [67], we try both. In addition, we tune the learning rate $\gamma$ extensively for all methods. We also try different gradient clipping, warmup steps, and weight-decay values. A summary of the hyperparameters tested and selected for each model size is in Appendix D. All the optimizers are described in depth in Appendix A.

# 4 Results

We structure our story starting with smaller models and batch sizes, and gradually scaling up to larger configurations. In some instances, we complement the core benchmarking results with additional ablations and possible best-practices.

## 4.1 Benchmarking at Small Scale: Training Models of 124M Parameters

**Using "small" batches.** We first report results when using batches of $32 \times 512$ tokens in Figure 3. We tune hyperparameters by training for 2.1B tokens (128k iterations) and then keep those hyperparameters for all other training durations. The best hyperparameters are reported in Appendix D.1. We observe how, for the smallest number of iterations we considered (1B tokens ≡ 64k), SOAP, ADOPT and AdEMAMix all outperform AdamW, with SOAP being the best. As we increase the number of iterations, AdEMAMix takes the lead while AdamW closes the gap with both ADOPT and SOAP. A sign-based methods such as Lion and Signum are expected to perform poorly when the batch size is small. Intuitively, this is due to the $\text{sign}(\cdot)$ operator being sensitive to gradient noise. As described in its original paper, MARS also performs poorly when the batch size is small. We found Prodigy, Muon and SF-AdamW to underperform in this setting compared to AdamW. On this scale, Prodigy suffers from the lack of bias correction of the learning rate, as well as being sensitive to $(\beta_1, \beta_2)$ (see Figure 17).

**Using "large" batches.** We now report results when using batches of $256 \times 512$ tokens — $8\times$ larger than for our small batch setting. Results in Figure 2 show how Signum, Mars, Lion, Prodigy greatly benefit from the increased batch size. Remarkably, we observe that the Prodigy method scales similarly to AdamW. We emphasize the possible community interest in this algorithm as its EMA Prodigy adaptively emulates the learning rate behaviour. For a small number of iterations (e.g. $T \in \{8\text{k}, 16\text{k}\}$ corresponding to 1B and 2B tokens), all methods outperform AdamW except for SF-AdamW and Sophia. As we increase the number of iterations ADOPT, SOAP, and AdEMAMix take the lead. In particular, AdEMAMix has a consistent lead over other methods. While we anticipated—in accordance with Vyas et al.[84]—that SOAP would greatly benefit from the larger batch size, its behavior remains relatively consistent compared to our previous small batch setting.

**Stability across training horizons.** As mentioned in Section 3, we tune hyperparameters training on 2.1B tokens and keep those hyperparameters when extending the training horizon. In Figure 3 we study whether it is possible to find better parameters for AdamW, SOAP, and AdEMAMix. When training on 16.8B tokens, we see it is beneficial to increase the $\beta_3$ from 0.999 to 0.9999. Without this improvement, SOAP ends up matching the performances of AdEMAMix when extending the training horizon further to 33.6B tokens ($\equiv$ 256k iterations). In our experiments, $\beta_3 = 0.999$ is only better than $\beta_3 = 0.9999$ when the number of training iterations is less than 32k. This matches observation

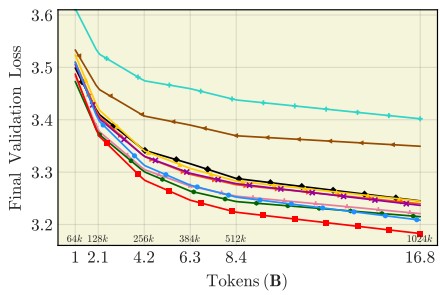
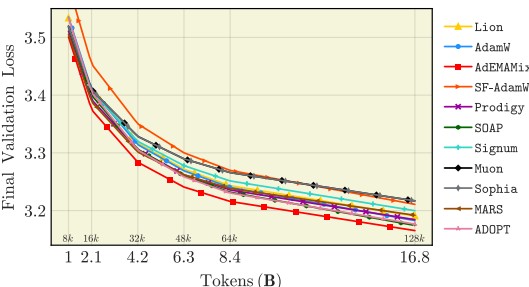

(a) Batch size $32 \times 512$ tokens.  (b) Batch size $256 \times 512$ tokens.

Figure 2: **Ranking of optimizers for** $124$**M models with small and large batch sizes.** In both **(a)** and **(b)**, we show the *final* validation loss for different training durations, corresponding to different numbers of tokens. Above each token number, we write the number of training iterations corresponding. In **(a)**, we use a "small" batch size of $32 \times 512$ tokens. In **(b)**, we use a larger batch size of $256 \times 512$ tokens.

from [62], which recommends reducing $\beta_3$ when training for fewer iterations. We also test whether the learning rate $\gamma$ changes as we increase the number of tokens/iterations. In Figure 5, we run a sweep over $\gamma$ when training for $16.8$**B** tokens. While for most methods, the best $\gamma$ obtained in the previous sweep remains optimal, this is not the case for SOAP and SF-AdamW, which can benefit from a larger $\gamma = 0.002$.

**WSD vs. cosine & linear $\gamma$-schedulers.** Learning rate schedulers received a lot of attention recently [79, 28]. We conducted a series of experiments comparing WSD [27, 90] and linear with cosine [49] learning rate schedulers. Surprisingly, the performance gap between these two schedulers observed in Figure 23 is often significant[2] for benchmarking optimizers. Consequently, we decided to adopt the cosine scheduler for all further experiments.

**Decaying $\gamma$ sufficiently.** In Figure 8 we show the impact of decaying more or less the learning rate $\gamma^{(t)}$. From $\gamma = 10^{-3}$ we train models using cosine decay down to $\gamma_{\text{end}} \in \{10^{-4}, 10^{-5}, \ldots, 10^{-9}\}$. We found that decaying the learning rate sufficiently matters. In particular, the often use rule consisting in decaying to $0.1 \times \gamma$ is suboptimal. This agrees with the recent works [28, 75, 4]. Building on this findings, we consistently use cosine decay down to $0.01 \times \gamma$.

> **Takeaway 1.** After the experiment in the small-batch setting, we conclude that: (i) AdEMAMix scales in the best manner with the number of iterations, SOAP underperforms AdamW when the length of training increases. ADOPT and Prodigy show almost equal performance across all training durations. Sign-based methods, predictably, underperform when the batch size is small, but what is interesting, is that Sophia diverges at all, even if trains with sufficiently small learning rate.

**Increasing the batch size further.** We also run an experiment with batches of $512 \times 512 = 262$**k** tokens, training for $64$**k** iterations. Results in Figure 3 show mostly consistent results. Noticeably MARS becomes the second best performing method behind AdEMAMix, followed closely by Prodigy, Lion, and SOAP. Interestingly, Signum performs comparably to AdamW.

> **Takeaway 2.** Taking into consideration large batch size setting, we found that many methods, once properly tuned, can show a remarkable performance compared to AdamW and also outperform it.

**Weight decay ablation.** As recent frameworks for LLM pretraining or ablation studies omit weight decay as a default non-zero hyperparameter, some setups even mislead by not incorporating weight decay in their experiments. In this work, we demonstrate the importance of weight decay and its impact across different optimizers. Surprisingly, increasing weight decay while keeping the learning

---

[2]We emphasize that the difference between the two schedulers is generally less than $5\%$ of the total compute spent. However, this still represents a significant gap in our benchmarking setup, e.g., SF-AdamW may outperform AdamW in some settings (see Figure 23).

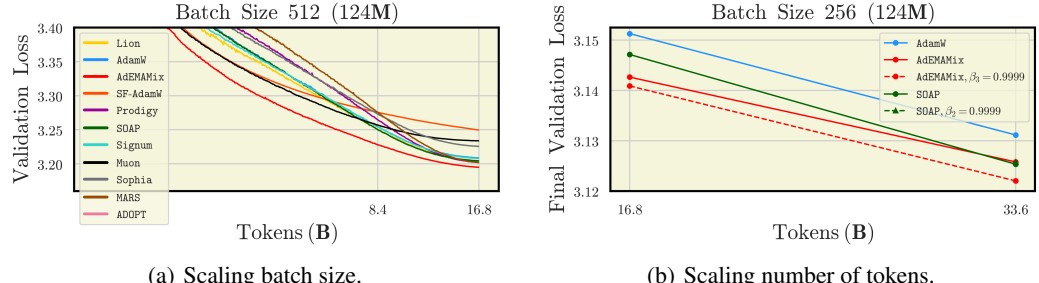

(a) Scaling batch size.  (b) Scaling number of tokens.

Figure 3: **Our results demonstrate that (a)**: scaling the batch size significantly improves `MARS`, `Signum`, `Lion` and `Prodigy` making them as good as `AdamW` even for a long training for $16.8\mathbf{B}$ tokens. Which was not the case in Figure 2 (b), where we still observed a significant gap in performance; and **(b)**: indeed, with scaling of the number of iterations, the gap between `SOAP` and `AdEMAMix` narrow and, finally, increases. But, on the other hand, with increase of the `AdEMAMix` $\beta_3$ parameter, the performance gap with `SOAP` reappears.

rate constant proves to be an effective technique for training on shorter horizons. This approach is so effective that methods like `Signum` and `Lion` with high weight decay significantly outperform `AdamW` without weight decay (see Figure 4). Implementation details also warrant attention. Coupled weight decay is still used in some settings, including the PyTorch [63] optimizer implementations. Notably, the popular implementation of `Signum` becomes ineffective when weight decay is applied. Highlighting this oversight for the community, we contribute by demonstrating our implementation of `Signum` (Algorithm 6) with decoupled weight decay. The influence of weight decay on model weights is intriguing. As is known, model weights typically grow during training, but weight decay, by modifying the optimized function, significantly reduces the growth of the model's parameter norm. Such ablations of weight decay are also of interest to the community [13, 40].

Regarding the ablation of weight decay for optimizers, we again select the best setup for each and conduct a sweep over weight decay values. Our results are presented in Figure 4 and in Figure 21.

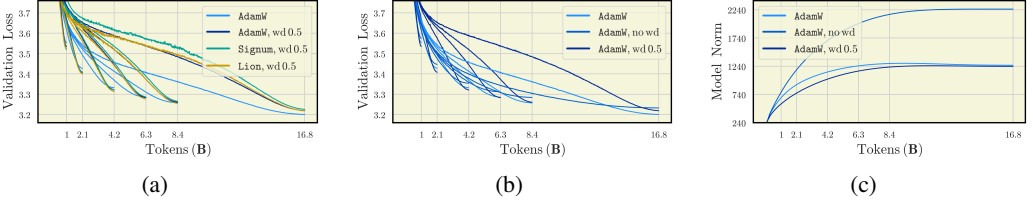

(a)  (b)  (c)

Figure 4: **Larger weight decay achieves significantly better results when training on fewer tokens.** In **(a)** we observe that runs of `AdamW`, `Signum`, and `Lion` with the large weight decay of $0.5$ consistently outperform the baseline `AdamW` with weight decay of $0.1$ for all training durations except for the last one. Notably, `Signum` and `Lion` with large weight decay perform even better than `AdamW` with the same learning rate. In **(b)**, we also consider a setting without weight decay. We observe that this is suboptimal not only for `AdamW`, but also for the majority of other optimizers (see Appendix E.2), while the typical weight decay of $0.1$ remains the best for large training durations. Importantly, in **(c)**, we ablate the impact od weight decay on the model's $\ell_2$ norm.

With our weight decay ablation, we are ready to provide one more insight.

> **Takeaway 3.** The use of weight decay, particularly a large decoupled weight decay term, can significantly impact the final loss value and optimizer behavior. However, for extended training horizons, a moderate, non-zero weight decay proves to be a robust option.

**Learning rate sensitivity.** Since we tune optimizers at a smaller scale and then extrapolate, we pose the question whether the best learning rate we have found so far transfers to the larger training duration. To verify this, we run $124\mathbf{M}$ model on $16.8\mathbf{B}$ tokens in $256 \times 512$ batch size setting, sweeping the learning rate across five typical values: $\{1e^{-4}, 3e^{-4}, 5e^{-4}, 1e^{-3}, 2e^{-3}\}$. The best

learning rate for each method at the moment of hyperparameter tuning on near Chinchilla-optimal 2.1B training duration we report in Appendix D.1. A summary of our results for larger number of tokens is provided in Figure 5 and detailed results of the sweep are presented in Appendix E.2.

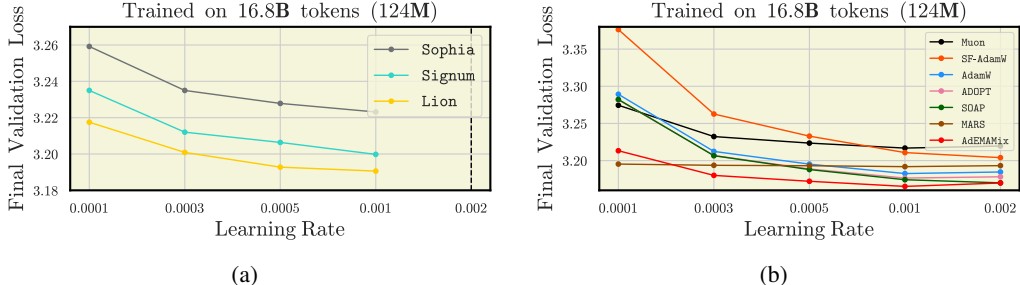

(a)                                        (b)

Figure 5: **Optimal learning rate stability across optimizers.** The optimal learning rate determined during tuning on 2.1B tokens remains consistent after a learning rate sweep on 16.8B tokens for most optimizers. In **(a)**, we observe that sign-based methods and similar to them `Sophia` diverge with increasing learning rate. Interestingly, in **(b)**, `SF-AdamW` and `SOAP` demonstrate the best performance with a large learning rate of 0.002. In our work, we further show that it is possible to increase the learning rate even more for such methods.

**Warmup ablation.** Another important ingredient of the pretraining is learning-rate warmup in the initial phase of training. Recent studies have explored the necessity of warmup in modern deep learning, with some investigating its elimination [39] and others ablating it to improve model performance and stability [92]. We focus on the latter, examining how warmup affects optimizer setup and whether it can significantly enhance performance. For each optimizer's best configuration, we vary warmup across three values: $\{0.27, 1, 4.2\}$B tokens, which corresponds to $\{2, 8, 32\}$k iterations. Our choice of the largest warmup value is inspired by [92]. We describe this experiment in Appendix E.2. Mainly, we observe that `Signum` and `SF-AdamW` perform better with a larger warmup of 8k steps when training on 16.8B tokens. We also ablate the claim from [92] that a warmup of 25% of the Chinchilla optimal duration is the best. However, our findings contradict this assertion (see Figure 18). We show that a moderate values of the warmup, generally, is better, however, different optimizers could prefer different number of warmup steps. As such, `SF-AdamW`, `Sophia`, `Signum` prefer larger warmup, which is clearly depicted in Figure 6.

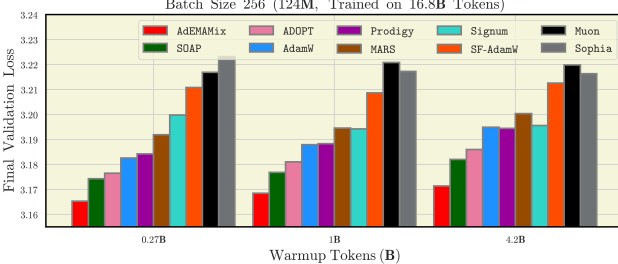

Figure 6: **Warmup ablation.** We report the final validation loss on the FineWeb dataset for 124M model trained on the batch size of 256. We sweep over the batch sizes of $\{1.56\%, 6.25\%, 25\%\}$ of the length of training, which corresponds to $\{2000, 8000, 32000\}$k iterations, respectively.

**Cosine vs WSD.** At the outset of our study, we indicated a preference for the cosine scheduler over WSD. In this section, we provide a more detailed ablation of this choice. Having optimally tuned the cosine scheduler for each optimizer, we replicate the setup of [28], which allows us to avoid adjusting additional hyperparameters. Our findings, which demonstrate the superiority of the cosine scheduler across various optimization methods, are presented in Figure 7, and in the Appendix Figures 23 and 24. These results not only validate our initial preference but also provide insights into the interaction between learning rate schedules and different optimizers in large-scale language model training.

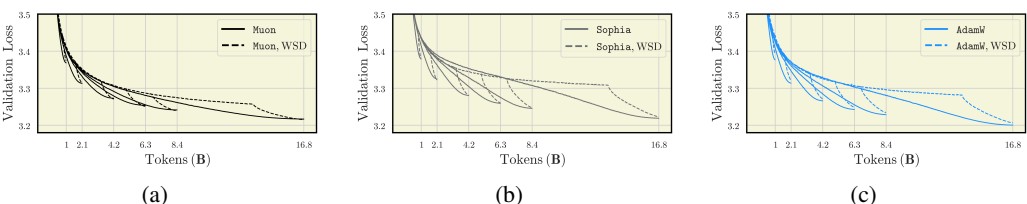

(a)                 (b)                 (c)

Figure 7: **Comparisons between WSD and cosine scheduler.** Notably, WSD and cosine scheduler behave differently with respect to optimizer. In **(a)**, the `Muon` optimizer shows a preference for WSD across most training durations. `Sophia` exhibits an almost perfect match between both schedulers. However, for `AdamW`, along with the majority of other optimizers studied (see Figure 24), we get a better performance with cosine. We also report a detailed comparison with linear scheduler in Appendix E.2.

## 4.2   Benchmarking at medium scale: Training Models of $210$M Parameters

In this section, we verify if our selected hyperparameters from smaller $124$M allow accurate transfer to a slightly larger model. We point out that the most important hyperparameters to be sweeped are learning rate and gradient clipping. Regarding the learning rate, we observe that it only becomes a sensitive choice for sign-based methods, while the optimal hyperparameters for `AdamW` remain the same.

**Results with a batch size of** $256 \times 512$**.** Results provided in the Appendix in Figure 20 are consistent with those obtained training $124$M models with large batches.

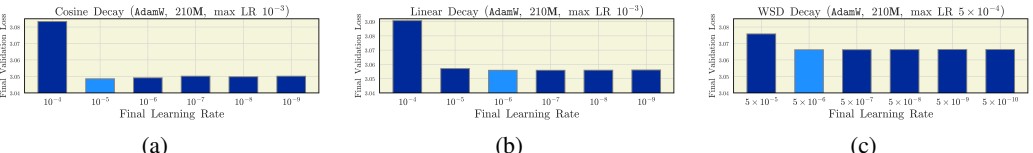

(a)                 (b)                 (c)

Figure 8: **Decaying the learning rate down to $0.01 \times \gamma_{\max}$ and beyond, instead of only to $10\%$** We observe a common pattern for different schedulers that decreasing the learning rate to moderate $10^{-2}$ value is a better choice than decreasing it down to zero. Interestingly, the linear learning rate scheduler for models at a given scale, requires $0.001 \times \gamma_{\max}$.

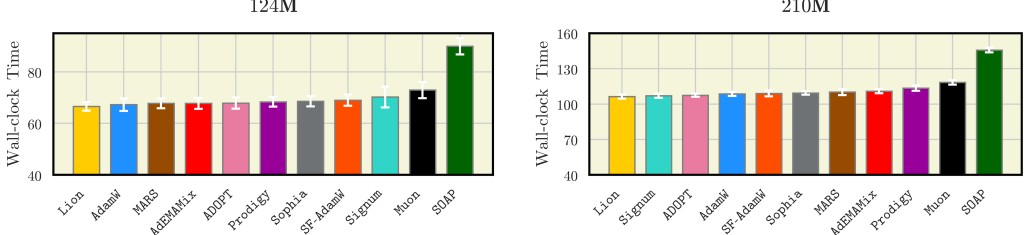

Figure 9: **Wall-clock time comparison.** After conducting experiments for $124$M and $210$M models, we are ready to present the wall-time comparison for each methods. For this purposes, we use a single GPU, and run each optimizer for 100 iterations on a small batch size without gradient accumulation and `torch.compile`. We report the wall-clock time per 100 iterations. We observe that all methods take the roughly the same time or very close time to complete 100 iterations, with the exception of `Muon` and `SOAP`. In addition, we point out that `SOAP`'s runtime exhibits a non-linear dependence on the model size, due to its preconditioner matrices operations which are fast only for certain matrices smaller than a predefined size.

## 4.3   Scaling Up: Benchmarking models of $583$M and $720$M Parameters

We pick three methods: AdamW, SOAP, and AdEMAMix, and run experiments with a larger model of $583$M parameters, and a large batch size of $2$M tokens. The goal being to get closer to one of the

settings described in [84]. We train for 6500 and 16000 iterations, corresponding to 13**B** and 32**B** tokens respectively.

**Comparison between our setting and [84].** We found several key differences between our codebase and [84]: (i) we decay the learning rate to $0.01 \times \gamma$ instead of $0.1 \times \gamma$, with $\gamma$ being the maximum learning rate, (ii) we use typical weight decay values of e.g. $0.1$ instead of smaller values such as $0.01$ or $0.0001$, (iii) we do not use a z-loss in addition to ours. It has been shown recently that properly decaying the learning rate has an important effect on the optimization [4]. We run an ablation to compare both settings and conclude that removing the z-loss and increasing the weight decay to $0.1$ improves the results. Results further improve when the learning rate is decayed more. This ablation is shown in Figure 8.

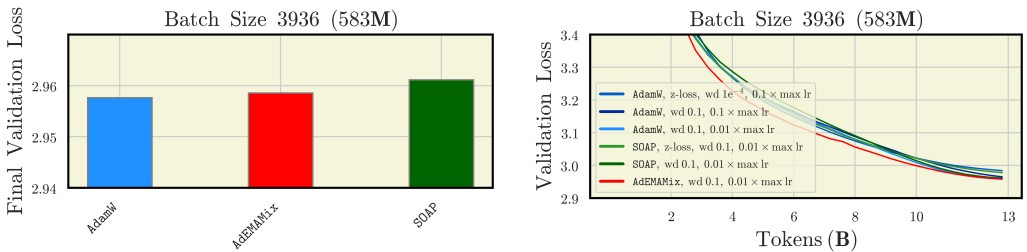

Figure 10: **Results for the** $583$**M model.** On the **left**, we show our results when training for $6500$ iterations. In this setting, AdamW gives best results, followed by AdEMAMix and then SOAP. This is surprising as it conflicts with findings from [84]. Those results are partly reconciled with the figure on the **right**. And we see that the difference in performance between models trained with and without the z-loss regularizer is quite minor.

## 5 Extension to MoEs

**Setup & Comparison.** Besides training dense Llama-like transformers, we also conver a comparison for MoE architectures [78]. Our variant of MoE is based on the Switch-Transformer implementation [20]. We use a classical linear gating with softmax and top-$k$ routing ($k = 2$) and $8$ experts. The activation functions remains the same as for the dense base model from Section 3. Such a configuration of the MoE model gives us approximately $520$**M** parameters. We cover additional details in Appendix E.6. In this setting we train with a batch size of $256 \times 512$ for $T \in \{42, 336\}$**k** iterations. Again we cover both a Chinchilla-optimal horizon and the beyond. We summarize the results in the following Table 1.

| **Opt.** | 42**k** | 336**k** |
|---|---|---|
| AdEMAMix | 22.37 | 18.47 |
| D-Muon | 22.67 | 18.51 |
| ADOPT | 22.70 | 18.58 |
| AdamW | 22.85 | 18.69 |
| Prodigy | 22.82 | 18.78 |

| **Opt.** | 42**k** | 336**k** |
|---|---|---|
| Lion | 23.20 | 18.87 |
| Signum | 23.31 | 19.09 |
| SF-AdamW | 23.34 | 19.13 |
| Sophia | 23.41 | 19.22 |
| MARS | 22.73 | 19.33 |

Table 1: **Final validation perplexity for MoE training** ($\downarrow$).

## 6 Discussion

**Our advices on tuning each method.** Overall, we validate the already widely used hyperparameters of $\lambda = 0.1$ and $T_{\text{warmup}} \approx 2$**k**. For `Lion`—as mentioned in [9]—we find that the best value for $\beta_1$ is consistently $0.99$. The mechanism for `Lion` seems similar to `AdEMAMix`, one can imagine that `Lion` could be better with larger $\beta_1$, which would require schedulers. We also pose an interesting observation toward `Prodigy`: while it may not be so efficient with a super small batch sizes, with scaling of the model size and the batch size it becomes almost as competitive as `AdamW`. Importantly, `Muon` and `D-Muon` performed poorly at a small scale with relatively small batch sizes (32, 256), however, as we see in Figure 1.

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
