$124\mathbf{M}$ model, we use batch sizes of $32 \times 512 = 16\mathbf{k}$, $256 \times 512 = 131\mathbf{k}$, and $512 \times 512 = 262\mathbf{k}$ tokens; for the $210\mathbf{M}$ model, we use a batch size of $256 \times 512 = 131\mathbf{k}$; and for $583\mathbf{M}$ model, we leverage the batch sizes of $1024 \times 512 = 524\mathbf{k}$ and $3936 \times 512 = 2\mathbf{M}$ tokens. Depending on the model size, we vary the number of iterations — also measured in tokens for compatibility with scaling laws and to accommodate different batch size settings. We train $124\mathbf{M}$ and $210\mathbf{M}$ models for equal durations of $\{1, 2.1, 4.2, 6.3, 8.4, 16.8\}\mathbf{B}$ tokens. This corresponds to $T \in \{64, 128, 256, 384, 512, 1024\}\mathbf{k}$ iterations for a batch size of 32, and $T \in \{8, 16, 32, 48, 64, 128\}\mathbf{k}$ iterations for a batch size of 256. For $583\mathbf{M}$ models, we train on $\{13, 32\}\mathbf{B}$ tokens, corresponding to $T \in \{6.5, 16\}\mathbf{k}$ iterations, resp. $T \in \{25, 61.5\}\mathbf{k}$ iterations, for a batch size of 3936, resp. 1024. In the setup with $720\mathbf{M}$ model, we have $T \in \{8, 16, 48\}\mathbf{k}$ iterations for a batch size of $1\mathbf{M}$

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

# Contents

# A    Optimizers we study

In this section, we describe all considered algorithms, presenting them in a unified formalism. We start with notation and then discuss the algorithms according to their logical grouping:

1. `Adam`-like methods: `AdamW` (Algorithm 1), `ADOPT` (Algorithm 2) and `AdEMAMix` (Algorithm 3).

2. Sign-based methods: `Lion` (Algorithm 4), `Signum` (Algorithms 5 and 6).

3. Approximate second-order optimizers: `Muon` (Algorithm 8), `SOAP` (Algorithm 10) and `Sophia` (Algorithm 11).

4. Learning rate / scheduler-free learning algorithms: `Schedule-Free AdamW` (Algorithm 12), `Prodigy` (Algorithm 13).

5. `MARS` algorithms: (Algorithms 14 to 16).

**Notation.** In our work, we denote all vectors and matrices using bold symbols, while non-bold symbols represent scalars. Let $\mathcal{L} : \mathcal{D} \to \mathbb{R}$ be an empirical loss function parameterized by $x$, and mapping batch of inputs $\xi \subset \mathcal{D}$ to $\mathbb{R}$. As $g = \nabla_x \mathcal{L}(x, \xi)$ we denote a stochastic gradient of the loss w.r.t. parameters $x$. For simplicity, we omit $x$ in $\nabla$ and write $\nabla \mathcal{L}(x, \xi)$. We use the following standardized notation for specific symbols in our work: batch size — $|\xi|$, learning rate — $\gamma$, weight decay — $\lambda$, momentum — $\beta$, iteration counter $t$ with the total number of iterations — $T$.

And basic notation for symbols in the algorithms: $m, v$ – are first and second moment estimates, respectively, with their bias corrected versions $\hat{m}, \hat{v}$, and beta parameters — $\beta_1, \beta_2$. We denote the dot product of two vectors $z, y$ as $\langle z, y \rangle$, while $z \odot y$ stands for their element-wise product. All division and addition operations in the described algorithms are element-wise.

### A.1 `AdamW`, `ADOPT`, `AdEMAMix`

`AdamW`. Our baseline — `Adam(W)`, has become a de facto optimizer for deep learning, demonstrating impressive performance across diverse domains: from tabular data to diffusion and language models.

The method originated from the ideas of `Adagrad` [19] and `RMSProp` [23], which utilize a second moment estimate $v$ in their update rule. However, `Adam(W)` enhanced this prior scheme by incorporating momentum [55, 82], establishing itself as a state-of-the-art method for a wide range of tasks. All other algorithms we consider also employ a similar, if not identical, momentum scheme.

Another key aspect of `AdamW` is its decoupled weight decay $\lambda$ [50], unlike `Adam`. We use the decoupled weight decay scheme for all methods to ensure consistency and emphasize its importance for optimizer comparison, hyperparameter tuning, and final performance. This is clearly observable, e.g., for `Signum` (Algorithm 5).

---

**Algorithm 1** `AdamW`

---

1: **Input:** Initial parameters $x_0$, number of iterations $T$, learning rate $\gamma_t$, weight decay $\lambda$, $\beta_1, \beta_2, \varepsilon$.
2: **Initialize:** $m_0 \leftarrow 0$, $v_0 \leftarrow 0$
3: **for** $t \in [T]$ **do**
4:     $g_t \leftarrow \nabla \mathcal{L}(x_t, \xi_t)$
5:     $m_t \leftarrow \beta_1 m_{t-1} + (1 - \beta_1) g_t$
6:     $v_t \leftarrow \beta_2 v_{t-1} + (1 - \beta_2) g_t \odot g_t$
7:     $\hat{m}_t \leftarrow m_t / (1 - \beta_1^t), \quad \hat{v}_t \leftarrow v_t / (1 - \beta_2^t)$
8:     $x_{t+1} \leftarrow x_t - \gamma_t \left( \frac{\hat{m}_t}{\sqrt{\hat{v}_t} + \varepsilon} + \lambda x_t \right)$
9: **end for**
10: **Return:** $x_T$

---

`ADOPT`. Recently, [83] proposed a modification of `Adam`, by removing the current gradient $g_t$ from the second moment estimate $v_t$ and changing the order of the momentum update $m_t$ and the normalization by the second moment estimate. As shown in `line 8` of Algorithm 2, the parameter update depends only on the previous value of the second moment estimate $v_{t-1}$. The authors analyze the convergence of `ADOPT` with the following update rule:

$$m_t \leftarrow \beta_1 m_{t-1} + (1 - \beta_1) \frac{g_t}{\max\{\sqrt{v_{t-1}}, \varepsilon\}},$$

$$x_{t+1} \leftarrow x_t - \gamma_t m_t.$$

However, the practical implementation differs in a few details. To tackle instabilities caused by near-zero gradients during the early stages of training, the authors propose using a clipping on $g_t / \max\{\sqrt{v_{t-1}}, \varepsilon\}$, which we formalize as the clamp operation. Given a vector $g$ and a positive scalar $c$, it is defined as:

$$\texttt{clamp}\,(g, c)^{(i)} = \min\left\{\max\left\{g^{(i)}, -c\right\}, c\right\}. \tag{1}$$

Thus, the element-wise clamp operation preserves $g_t$ from the division by near-zero values.

The authors theoretically claim that ADOPT achieves the optimal convergence bound for non-convex objectives, regardless of the choice of the $\beta_2$ parameter. We empirically investigate this claim and observe that, contrary to the theoretical results, there is a significant performance gap for different choices of $\beta_2$ in practice.

---

**Algorithm 2** ADOPT

1: **Input:** Initial parameters $x_0$, number of iterations $T$, learning rate $\gamma_t$, weight decay $\lambda$, $\beta_1, \beta_2, \varepsilon$.
2: **Initialize:** $m_0 \leftarrow 0$, $v_0 \leftarrow \nabla\mathcal{L}(x_0, \xi_0) \odot \nabla\mathcal{L}(x_0, \xi_0)$
3: **for** $t \in [T]$ **do**
4:      $g_t \leftarrow \nabla\mathcal{L}(x_t, \xi_t)$
5:      $c_t \leftarrow t^{1/4}$                               ▷ Update clipping value schedule
6:      $m_t \leftarrow \beta_1 m_{t-1} + (1 - \beta_1)\texttt{clamp}\left(\frac{g_t}{\max\{\sqrt{v_{t-1}}, \varepsilon\}}, c_t\right)$
7:      $v_t \leftarrow \beta_2 v_{t-1} + (1 - \beta_2)g_t \odot g_t$
8:      $x_{t+1} \leftarrow x_t - \gamma_t (m_t + \lambda x_t)$               ▷ Update without $v_t$
9: **end for**
10: **Return:** $x_T$

---

**AdEMAMix.** Another Adam-like optimizer we study is AdEMAMix [62]. This work argues that using a single EMA to accumulate past gradients in the first moment estimate $m$ can be sub-optimal, as it cannot simultaneously prioritize both immediate past and older gradients. In Algorithm 3, the authors incorporate two EMAs: one — Adam-like EMA for $m$ (fast), and another — a slow EMA $m^{\text{slow}}$ (see line 7) with an additional $\beta_3$ parameter. In its update rule, the algorithms balances fast and slow EMAs with the constant factor $\alpha$ (see line 10 of Algorithm 3). This algorithmic design helps AdEMAMix benefit from older gradients and results in smoother loss curves during training.

However, to mitigate the effect of early instabilities, the authors use two additional schedulers for $\alpha$ and $\beta_3$ – alpha_scheduler and beta_scheduler, formalized in our work as follows:

$$\texttt{alpha\_scheduler}(t, \alpha, T_\alpha) = \min\left\{\frac{t\alpha}{T_\alpha}, \alpha\right\},$$

$$\texttt{beta\_scheduler}(t, \beta_3, \beta_{\text{start}}, T_{\beta_3}) = \min\left\{\exp\left(\frac{\log(\beta_{\text{start}})\log(\beta_3)}{\left(1 - \frac{t}{T_{\beta_3}}\right)\log(\beta_3) + \frac{t}{T_{\beta_3}}\log(\beta_{\text{start}})}\right), \beta_3\right\}.$$

In all experiments, we set $\beta_{\text{start}} = \beta_1$, and the warmup parameters equal to the length of training: $T_\alpha = T_{\beta_3} = T$.

One thing that should be commented — $\alpha$ and $\beta$ schedulers are seemingly contradict the idea of WSD scheduler. However, setting $T_\alpha, T_{\beta_3}$ to be longer than the first checkpoint of the WSD does not significantly impact the final performance. Thus, AdEMAMix can still be combined with recent findings regarding the WSD scheduler.

## A.2 Sign-based methods: Lion and Signum

Another branch of methods includes sign-based Lion and Signum. To some extend, one can classify Adam as a sign-based method also, but we mention only Lion and Signum as they explicitly incorporate the sign operation in the update rule.

These methods, particularly Signum, have been unfairly overlooked in the context of LLM pretraining. However, our results demonstrate that, with sufficiently large batch sizes and at moderate model scales, these optimizers perform on par with Adam, and in some cases, even outperform it.

---

**Algorithm 3** AdEMAMix

---

1: **Input:** Initial parameters $\boldsymbol{x}_0$, number of iterations $T$, learning rate $\gamma_t$, weight decay $\lambda$, $\beta_1$, $\beta_2$, $\beta_3$, $\beta_{\text{start}}$, $\alpha$, beta_scheduler, alpha_scheduler, warmup parameters $T_{\beta_3}$ and $T_\alpha$, $\varepsilon$.

2: **Initialize:** $\boldsymbol{m}_0 \leftarrow \boldsymbol{0}$, $\boldsymbol{m}_0^{\text{slow}} \leftarrow \boldsymbol{0}$, $\boldsymbol{v}_0 \leftarrow \boldsymbol{0}$

3: **for** $t \in [T]$ **do**

4:     $\beta_3(t) \leftarrow$ beta_scheduler$(t, \beta_3, \beta_{\text{start}}, T_{\beta_3})$, $\alpha(t) \leftarrow$ alpha_scheduler$(t, \alpha, T_\alpha)$    ▷ Update $\beta_3$ and $\alpha$ schedulers

5:     $\boldsymbol{g}_t \leftarrow \nabla\mathcal{L}(\boldsymbol{x}_t, \boldsymbol{\xi}_t)$

6:     $\boldsymbol{m}_t \leftarrow \beta_1 \boldsymbol{m}_{t-1} + (1 - \beta_1)\boldsymbol{g}_t$

7:     $\boldsymbol{m}_t^{\text{slow}} \leftarrow \beta_3(t)\boldsymbol{m}_{t-1}^{\text{slow}} + (1 - \beta_3(t))\boldsymbol{g}_t$                      ▷ Update slow EMA

8:     $\boldsymbol{v}_t \leftarrow \beta_2 \boldsymbol{v}_{t-1} + (1 - \beta_2)\boldsymbol{g}_t \odot \boldsymbol{g}_t$

9:     $\hat{\boldsymbol{m}}_t \leftarrow \boldsymbol{m}_t/(1 - \beta_1^t)$,   $\hat{\boldsymbol{v}}_t \leftarrow \boldsymbol{v}_t/(1 - \beta_2^t)$

10:    $\boldsymbol{x}_{t+1} \leftarrow \boldsymbol{x}_t - \gamma_t \left( \frac{\hat{\boldsymbol{m}}_t + \alpha(t)\boldsymbol{m}_t^{\text{slow}}}{\sqrt{\hat{\boldsymbol{v}}_t} + \varepsilon} + \lambda \boldsymbol{x}_t \right)$

11: **end for**

12: **Return:** $\boldsymbol{x}_T$

---

953   Lion.   And the first method of this kind is Lion [9]. This optimizer is symbolically discovered in
954   the program space of first-order optimization primitives. Lion updates its EMA of $\boldsymbol{m}$ after updating
955   the parameters and has additional term $(1 - \beta_1)\boldsymbol{g}$ whis adds to the momentum. This interpolation
956   $\beta_1 \boldsymbol{m}_{t-1} + (1 - \beta_1)\boldsymbol{g}_t$ (see line 6 of Algorithm 4) makes the symbolic-discovered idea behind
957   Lion similar to the idea of the AdEMAMix optimizer.

---

**Algorithm 4** Lion

---

1: **Input:** Initial parameters $\boldsymbol{x}_0$, number of iterations $T$, learning rate $\gamma_t$, weight decay $\lambda$, $\beta_1$, $\beta_2$.

2: **Initialize:** $\boldsymbol{m}_0 \leftarrow \boldsymbol{0}$

3: **for** $t \in [T]$ **do**

4:     $\boldsymbol{g}_t \leftarrow \nabla\mathcal{L}(\boldsymbol{x}_t, \boldsymbol{\xi}_t)$

5:     $\boldsymbol{m}_t \leftarrow \beta_2 \boldsymbol{m}_{t-1} + (1 - \beta_2)\boldsymbol{g}_t$                          ▷ Update EMA of $\boldsymbol{g}_t$

6:     $\boldsymbol{x}_{t+1} \leftarrow \boldsymbol{x}_t - \gamma_t \left( \text{sign}\left(\beta_1 \boldsymbol{m}_{t-1} + (1 - \beta_1)\boldsymbol{g}_t\right) + \lambda \boldsymbol{x}_t \right)$

7: **end for**

8: **Return:** $\boldsymbol{x}_T$

---

958   Signum.   Another sign-based method, which is the adoption of signSGD — Signum [6], or
959   signSGD with momentum. This method differs from Lion in the interpolation term between EMA
960   of momentum and current gradient, as well as in the Signum's update rule a current EMA is used.

961   Importantly, while Signum is not as widespread for LLM pretraining and stands mostly as a theoretical
962   artifact, recent practitioner's studies also start to use Signum for scalable training [96]. Mostly, due
963   to the memory-efficiency of Signum compared to AdamW.

964   In this regard, we want to make an important point — many recent PyTorch [63] implementations of
965   the Signum optimizer, unlikely, are not suitable for this method, impairing the potential performance
966   from using it.

967   The main problem of many open-source implementations is a use of decoupled weight decay in
968   the PyTorch implementation of SGDM (SGD with momentum) [82]. Indeed, with a decoupled weight
969   decay, the update of Algorithm 5 transforms into:

$$\boldsymbol{x}_{t+1} \leftarrow \boldsymbol{x}_t - \gamma_t \text{sign}\left(\beta \boldsymbol{m}_{t-1} + (1 - \beta)\boldsymbol{g}_t - \lambda(1 - \beta)\boldsymbol{g}_t\right),$$

970   which affects the sign of the update, leading to potentially wrong optimization direction if the weight
971   decay is sufficiently large.

972   Another popular failure while using Signum is incorrectly tractable PyTorch implementation of
973   SGDM. It does not include such EMA as line 5 in Algorithm 5, on the other hand, in PyTorch, the
974   momentum update depends on the dampening parameter $\tau$:

$$\boldsymbol{m}_t \leftarrow \beta \boldsymbol{m}_{t-1} + (1 - \tau)\boldsymbol{g}_t,$$

which is zero by default. Therefore, the typical update rule, reflecting the actual `Signum` behavior in practice corresponds to the following update:

$$\boldsymbol{x}_{t+1} \leftarrow \boldsymbol{x}_t - \gamma_t \left( \text{sign} \left( \beta \boldsymbol{m}_{t-1} + (1 - \tau) \boldsymbol{g}_t \right) + \lambda \boldsymbol{x}_t \right),$$

where the weight decay is decoupled and, consequently, does not affect the `sign`.

$$\boldsymbol{g}_t \leftarrow \boldsymbol{g}_t + \beta \boldsymbol{m}_t,$$

improves `Signum`. Since enabling Nesterov momentum requires zero dampening $\tau$, we revisited the description of Algorithm 5, and propose more practical, PyTorch-compatible version of `Signum` in Algorithm 6.

---

**Algorithm 5** `Signum` (basic)

1: **Input:** Initial parameters $\boldsymbol{x}_0$, number of iterations $T$, learning rate $\gamma_t$, weight decay $\lambda$, momentum $\beta$.
2: **Initialize:** $\boldsymbol{m}_0 \leftarrow \boldsymbol{0}$
3: **for** $t \in [T]$ **do**
4: $\quad \boldsymbol{g}_t \leftarrow \nabla \mathcal{L}(\boldsymbol{x}_t, \boldsymbol{\xi}_t)$
5: $\quad \boldsymbol{m}_t \leftarrow \beta \boldsymbol{m}_{t-1} + (1 - \beta) \boldsymbol{g}_t$
6: $\quad \boldsymbol{x}_{t+1} \leftarrow \boldsymbol{x}_t - \gamma_t \left( \text{sign} \left( \boldsymbol{m}_t \right) + \lambda \boldsymbol{x}_t \right)$
7: **end for**
8: **Return:** $\boldsymbol{x}_T$

---

**Algorithm 6** `Signum` (our PyTorch variant)

1: **Input:** Initial parameters $\boldsymbol{x}_0$, number of iterations $T$, learning rate $\gamma_t$, weight decay $\lambda$, momentum $\beta$.
2: **Initialize:** $\boldsymbol{m}_0 \leftarrow \boldsymbol{0}$
3: **for** $t \in [T]$ **do**
4: $\quad \boldsymbol{g}_t \leftarrow \mathcal{L}(\boldsymbol{x}_t, \boldsymbol{\xi}_t)$
5: $\quad \boldsymbol{m}_t \leftarrow \beta \boldsymbol{m}_{t-1} + \boldsymbol{g}_t$
6: $\quad \boldsymbol{x}_{t+1} \leftarrow \boldsymbol{x}_t - \gamma_t \left( \text{sign} \left( \beta \boldsymbol{m}_t + \boldsymbol{g}_t \right) + \lambda \boldsymbol{x}_t \right)$
7: **end for**
8: **Return:** $\boldsymbol{x}_T$

---

Moreover, to prevent other researchers and practitioners from the possible wrong use of `Signum` and for the reproducibility purposes, we provide our Python code.

Listing 1: `Signum` code skeleton using PyTorch

```python
from typing import Dict

import torch

class Signum(torch.optim.Optimizer):
    def __init__(
        self,
        params,
        lr=1e-3,
        momentum=0,
        dampening=0,
        weight_decay=0,
        nesterov=False,
        sign_update=True,
    ):
        if lr < 0.0:
            raise ValueError(f"Invalid learning rate: {lr}")
        if momentum < 0.0:
            raise ValueError(f"Invalid momentum value: {momentum}"
                )
        if weight_decay < 0.0:
            raise ValueError(f"Invalid weight_decay value: {
                weight_decay}")

        defaults = dict(
                lr=lr,
```

```python
                momentum=momentum,
                dampening=dampening,
                weight_decay=weight_decay,
                nesterov=nesterov,
                sign_update=sign_update,
            )
            if nesterov and (momentum <= 0 or dampening != 0):
                raise ValueError("Nesterov momentum requires a
                    momentum and zero dampening")
            super().__init__(params, defaults)

    def __setstate__(self, state):
        super().__setstate__(state)
        for group in self.param_groups:
            group.setdefault("nesterov", False)

    @torch.no_grad()
    def _init_state(self, example, state=None):
        assert isinstance(example, torch.Tensor)
        assert isinstance(state, Dict) or state is None
        if state is None:
            state = {}
        state["step"] = 0
        state["momentum_buffer"] = torch.clone(example).detach()
        return state

    @torch.no_grad()
    def _compute_update(
        self, grad, state, lr, momentum, nesterov, dampening,
            sign_update, **kwargs
    ):
        if momentum != 0:  # Signum check
            buf = state["momentum_buffer"]
            buf.mul_(momentum).add_(grad, alpha=1 - dampening)

            if nesterov:
                grad = grad.add(buf, alpha=momentum)
            else:
                grad = buf

        if sign_update:
            grad = grad.sign()

        return grad * (-lr)

    @torch.no_grad()
    def step(self, closure=None):
        """Performs a single optimization step.

        Args:
            closure (Callable, optional): A closure that
                reevaluates the model
                and returns the loss.
        """
        loss = None
        if closure is not None:
            with torch.enable_grad():
                loss = closure()
```

```
1069        for group in self.param_groups:
1070            for p in group["params"]:
1071                if p.grad is None:
1072                    continue
1073
1074                grad = p.grad
1075                state = self.state[p]
1076
1077                if group["weight_decay"] != 0:
1078                    p.mul_(1 - group["lr"] * group["weight_decay"
1079                        ])
1080
1081                if len(state) == 0:
1082                    self._init_state(example=p, state=state)
1083                    if not group["momentum"]:
1084                        state.pop("momentum_buffer", None)
1085
1086                state["step"] += 1
1087
1088                update = self._compute_update(
1089                    grad,
1090                    state,
1091                    group["lr"],
1092                    group["momentum"],
1093                    group["nesterov"],
1094                    group["dampening"],
1095                    group["sign_update"],
1096                )
1097
1098                p.add_(update)
1099
1100        return loss
```

## A.3 Muon, SOAP, Sophia

Next page of the methods covers algorithms that rather aim to use more information about the problem's curvature (SOAP [84], Sophia [46]) or perform fast updates of matrix parameters involving higher order computations (Muon [33]).

Contrary to the chronological order, we discuss them starting from the recent one — Muon and end up with Sophia.

**Muon.** Specifically designed for the speedrun comparisons, this method surpasses AdamW baseline on nanoGPT pretraining benchmark [32]. Claims from the Muon extend to faster learning, lower memory usage and better sample-efficiency with a small wall-clock time overhead. However, there are not much to say about the final performance given a particular budget of tokens to train on.

The reason why the Muon is a good option for speedrun pretraining lies in its structure — Muon treats different layers in different way. One dimensional (1D) parameters, large embedding layers, scalar parameters (such as Layer Norm parameters) and the output layer of LLM (lm_head) are optimized by AdamW. And all parameters with two or more dimensions, e.g., Multi-Head Attention layers are optimized by Algorithm 7, which we call MuonNon1D.

Inspired by Shampoo's preconditioners [24], the authors of MuonNon1D incorporated an orthogonalization step to compute the SVD transformation of gradient matrix. Before the orthogonalization step, MuonNon1D represents SGD with Nesterov momentum. To ensure fast orthogonalization procedure, the authors, insiped by [5], use Newton-Schulz procedure [25]. As the number of Newton-Schulz iterations increases, the closer resulting matrix becomes to $UV^\top$ from SVD transformation. The authors also mention that Muon can be thought of as a second way of smoothing spectral steepest descent [8], with a different set of memory and runtime trade-offs compared to Shampoo.

---

**Algorithm 7** MuonNon1D (for non-1D parameters)

---

1: **Input:** Initial non-1D parameters $\boldsymbol{x}_0$, number of iterations $T$, learning rate $\gamma_t$, momentum $\beta$, number of Newton-Schulz iterations $T_{\text{NS}}$, $a$, $b$, $c$ coefficients.
2: **Initialize:** $\boldsymbol{m}_0 \leftarrow \boldsymbol{0}$
3: **for** $t \in [T]$ **do**
4: $\quad \boldsymbol{g}_t \leftarrow \nabla\mathcal{L}(\boldsymbol{x}_t, \boldsymbol{\xi}_t)$
5: $\quad \boldsymbol{m}_t \leftarrow \beta\boldsymbol{m}_{t-1} + \boldsymbol{g}_t$
6: $\quad \boldsymbol{g}_t \leftarrow \beta\boldsymbol{m}_t + \boldsymbol{g}_t$ $\qquad\qquad\qquad\qquad$ ▷ Practical implementation of Nesterov momentum
7: $\quad$ **Set:** $\boldsymbol{w}_0 \leftarrow \boldsymbol{g}_t/\|\boldsymbol{g}_t\|_F$
8: $\quad$ **for** $n \in [T_{\text{NS}}]$ **do**
9: $\qquad \boldsymbol{w}_{n+1} \leftarrow a\boldsymbol{w}_n + b\boldsymbol{w}_n\boldsymbol{w}_n^\top + c\left(\boldsymbol{w}_n\boldsymbol{w}_n^\top\right)^2\boldsymbol{w}_n$ $\qquad$ ▷ Newton-Schulz iteration
10: $\quad$ **end for**
11: $\quad \boldsymbol{x}_{t+1} \leftarrow \boldsymbol{x}_t - \gamma_t\boldsymbol{w}_{T_{\text{NS}}}$
12: **end for**
13: **Return:** $\boldsymbol{x}_T$

---

---

**Algorithm 8** Muon (general scheme)

---

1: **Input:** Initial parameters $\boldsymbol{x}_0$, number of iterations $T$. Muon's parameters: learning rate $\gamma_t^{\text{M}}$, momentum $\beta$, number of Newton-Schulz iterations $T_{\text{NS}}$, $a$, $b$, $c$ coefficients. AdamW's parameters: learning rate $\gamma_t^{\text{A}}$, weight decay $\lambda$, $\beta_1$, $\beta_2$, $\varepsilon$.
2: **for** $t \in [T]$ **do** $\boldsymbol{x}_t \in \{\texttt{embeds}, \texttt{scalar\_params}, \texttt{lm\_head}\}$
3: $\quad \boldsymbol{x}_t^{\text{A}} \leftarrow \boldsymbol{x}_t$
4: $\quad \boldsymbol{x}_{t+1}^{\text{A}} \leftarrow \texttt{AdamW}\left(\boldsymbol{x}_t^{\text{A}}, \gamma_t^{\text{A}}, \lambda, \beta_1, \beta_2, \varepsilon, T=1\right)$ $\qquad\qquad$ ▷ One iteration of AdamW
5: $\quad \boldsymbol{x}_t^{\text{M}} \leftarrow \boldsymbol{x}_t$
6: $\quad \boldsymbol{x}_{t+1}^{\text{M}} \leftarrow \texttt{MuonNon1D}\left(\boldsymbol{x}_t^{\text{M}}, \gamma_t^{\text{M}}, T_{\text{NS}}, \beta, a, b, c, T=1\right)$ $\qquad$ ▷ One iteration of MuonNon1D
7: **end for**
8: **Return:** $\boldsymbol{x}_T^{\text{A}}, \boldsymbol{x}_T^{\text{M}}$

---

Importantly, we noticed that the original algorithmic description of Muon optimizer, provided in the official repository[3], differs from the actual one, presented in Algorithm 7. In the original code, as well as in our benchmarking, weight decay do not applies to the matrix parameters in the optimizer state of MuonNon1D, which means that the only weight decay used during training is AdamW's weight decay. From this perspective, we observe that the gap between the final loss values for runs with $0.1$ and $0$ weight decay values almost disappears, while the run with $0.5$ weight decay becomes the worst, which is not the case for other optimizers. We describe this in our weight decay ablations.

SOAP. [84] proposed new, improved modification of Shampoo [24]. SOAP reduces the computational overhead optimizing only two dimensional layers (2D) via Algorithm 9, while running AdamW for 1D layers. At initialization, preconditioners are computed via eigenvector decomposition of the initial gradient matrices eigenbasis $\left(\nabla\mathcal{L}(\boldsymbol{x}_0, \boldsymbol{\xi}_0)\nabla\mathcal{L}(\boldsymbol{x}_0, \boldsymbol{\xi}_0)^\top\right)$: $\nabla\mathcal{L}(\boldsymbol{x}_0, \boldsymbol{\xi}_0)\nabla\mathcal{L}(\boldsymbol{x}_0, \boldsymbol{\xi}_0)^\top = \boldsymbol{q}\Lambda\boldsymbol{q}^{-1}$, where $\Lambda$ stands for the diagonal matrix whose diagonal elements are the corresponding eigenvalues. For the rest of the iterations, SOAPNon1D performs the QR decomposition (see lines 15, 16 of Algorithm 9) for all 2D layers, which is the main computational part of the method.

A key idea behind the SOAP optimizer is:

1. Given the slowly changing coordinate basis provided by eigenvectors $\boldsymbol{l}$ and $\boldsymbol{r}$, SOAP updates its second moment estimates in this basis, i.e., it runs AdamW in another, rotated space.

2. To update the eigenvectors of $\boldsymbol{l}$ and $\boldsymbol{r}$, SOAP runs QR decomposition with preconditioning frequency $\phi$.

In Algorithm 9, if one would set both $\boldsymbol{q}_l$ and $\boldsymbol{q}_r$ to identity, then we would recover AdamW.

The overall SOAP algorithm can be formalized as follows:

---

[3] https://github.com/KellerJordan/modded-nanogpt

---

**Algorithm 9** `SOAPNon1D` (for non-1D parameters)

1: **Input:** Initial parameters $\boldsymbol{x}_0$, number of iterations $T$, learning rate $\gamma_t$, weight decay $\lambda$, $\beta_1$, $\beta_2$, preconditioning frequency $\phi$, $\varepsilon$.
2: **Initialize:** $\boldsymbol{m}_0 \leftarrow \boldsymbol{0}$, $\boldsymbol{v}_0 \leftarrow \boldsymbol{0}$
3: **Initialize preconditioners:** $\boldsymbol{q}_l, \boldsymbol{q}_r \leftarrow \texttt{eigenbasis}\left(\nabla\mathcal{L}(\boldsymbol{x}_0, \boldsymbol{\xi}_0)\nabla\mathcal{L}(\boldsymbol{x}_0, \boldsymbol{\xi}_0)^\top\right)$
4: **for** $t \in [T]$ **do**
5:     $\boldsymbol{g}_t \leftarrow \nabla\mathcal{L}(\boldsymbol{x}_t, \boldsymbol{\xi}_t)$
6:     $\boldsymbol{g}'_t \leftarrow \boldsymbol{q}_l^\top \boldsymbol{g}_t \boldsymbol{q}_r$                                                 $\triangleright$ Rotate $\boldsymbol{g}_t$
7:     $\boldsymbol{m} \leftarrow \beta_1 \boldsymbol{m}_{t-1} + (1-\beta_1)\boldsymbol{g}_t$
8:     $\boldsymbol{m}'_t \leftarrow \boldsymbol{q}_l^\top \boldsymbol{m}_t \boldsymbol{q}_r$                      $\triangleright$ Compute `Adam`'s statistics in rotational space
9:     $\boldsymbol{v}_t \leftarrow \beta_2 \boldsymbol{v}_{t-1} + (1-\beta_2)\boldsymbol{g}'_t \odot \boldsymbol{g}'_t$
10:    $\gamma_t \leftarrow \gamma_t \frac{\sqrt{1-\beta_2^t}}{1-\beta_1^t}$                              $\triangleright$ Optional: use bias correction
11:    $\boldsymbol{x}_{t+1} \leftarrow \boldsymbol{x}_t - \gamma_t\left(\boldsymbol{q}_l \frac{\boldsymbol{m}'_t}{\sqrt{\boldsymbol{v}_t}+\varepsilon}\boldsymbol{q}_r^\top + \lambda\boldsymbol{x}_t\right)$      $\triangleright$ Perform update in original space
12:    $\boldsymbol{l}_t \leftarrow \beta_2\boldsymbol{l}_{t-1} + (1-\beta_2)\boldsymbol{g}_t\boldsymbol{g}_t^\top$                $\triangleright$ Update preconditioners
13:    $\boldsymbol{r}_t \leftarrow \beta_2\boldsymbol{r}_{t-1} + (1-\beta_2)\boldsymbol{g}_t^\top\boldsymbol{g}_t \; t \equiv 1 \pmod{\phi}$
14:    $\boldsymbol{q}_l \leftarrow \texttt{QR}\left(\boldsymbol{l}_t\boldsymbol{q}_l\right)$
15:    $\boldsymbol{q}_r \leftarrow \texttt{QR}\left(\boldsymbol{r}_t\boldsymbol{q}_r\right)$
16: **end for**
17: **Return:** $\boldsymbol{x}_T$

---

---

**Algorithm 10** `SOAP` (general scheme)

1: **Input:** Initial parameters $\boldsymbol{x}_0$, number of iterations $T$, learning rate $\gamma_t$, weight decay $\lambda$, $\beta_1$, $\beta_2$, preconditioning frequency $\phi$, $\varepsilon$.
2: **for** $t \in [T]$ **do** $\boldsymbol{x}_t \in \{\texttt{embeds}, \texttt{scalar\_params}, \texttt{lm\_head}\}$
3:     $\boldsymbol{x}_t^\texttt{A} \leftarrow \boldsymbol{x}_t$
4:     $\boldsymbol{x}_{t+1}^\texttt{A} \leftarrow \texttt{AdamW}\left(\boldsymbol{x}_t^\texttt{A}, \gamma_t, \lambda, \beta_1, \beta_2, \varepsilon, T=1\right)$            $\triangleright$ One iteration of `AdamW`
5:     $\boldsymbol{x}_t^\texttt{S} \leftarrow \boldsymbol{x}_t$
6:     $\boldsymbol{x}_{t+1}^\texttt{S} \leftarrow \texttt{SOAPNon1D}\left(\boldsymbol{x}_t^\texttt{S}, \gamma_t, \lambda, \beta_1, \beta_2, \varepsilon, T=1\right)$      $\triangleright$ One iteration of `SOAPNon1D`
7: **end for**
8: **Return:** $\boldsymbol{x}_T^\texttt{A}, \boldsymbol{x}_T^\texttt{S}$

---

**Sophia.** Despite being named as second-order optimizer, `Sophia` [46] performs an update, quite similar to `Adam`'s. It also leverages the diagonal preconditioner $\boldsymbol{h}$, but not the curvature information of the optimization problem, which depends on the non-diagonal terms of the Hessian. One should notice that `Sophia` were introduced with two types of preconditioners — Hutchinson [3] and Gauss-Newton-Bartlett [51]. Since the latter one shows more promising performance, we consider only this type of preconditioner for `Sophia`.

Every $\phi$ iterations, `Sophia` updates its second moment estimate by computing the gradient $\hat{\boldsymbol{g}}$ of the empirical loss $\mathcal{L}$ given `softmax` of the logits instead of the true logits. Multiplying by the batch size, we obtain $\hat{\boldsymbol{h}}$, after that, `Sophia` updates the EMA of $\hat{\boldsymbol{h}}$.

Importantly, we found out, that algorithmic description of `Sophia` in the original paper differs in minor details from the code implementation[4]. Indeed, the update rule in their work formulates as follows:

$$\boldsymbol{x}_{t+1} \leftarrow \boldsymbol{x}_t - \gamma_t\texttt{clamp}\left(\frac{\boldsymbol{m}_t}{\max\{\rho\boldsymbol{h}_t, \varepsilon\}}, 1\right),$$

where `clamp` is defined as in Equation (1). On the other hand, the code from the official repository suggests:

Listing 2: `Sophia` update skeleton using PyTorch

```
# update step
```

---
[4]https://github.com/Liuhong99/Sophia

```
1159    step_t += 1
1160
1161    # Perform stepweight decay
1162    param.mul_(1 - lr * weight_decay)
1163
1164    # Decay the first and second moment running average coefficient
1165    exp_avg.mul_(beta1).add_(grad, alpha=1 - beta1)
1166
1167  else:
1168      step_size_neg = -lr
1169
1170      ratio = (exp_avg.abs() / (rho * bs * hess + 1e-15)).clamp(None
1171          , 1)
1172      param.addcmul_(exp_avg.sign(), ratio, value=step_size_neg)
```

Therefore, the actual update of `Sophia` is wrongly tractated in the original paper and should be corrected and equal to the `line 16` of Algorithm 11.

> **Takeaway 4.** *The actual update rule of `Sophia` differs from its description in the original paper.*

---

**Algorithm 11** `Sophia`

---

1: **Input:** Initial parameters $x_0$, number of iterations $T$, learning rate $\gamma_t$, weight decay $\lambda$, $\beta_1$, $\beta_2$, estimator frequency $\phi$, scaling factor $\rho$, $\varepsilon$.
2: **Initialize:** $m_0 \leftarrow 0$, $h_0 \leftarrow 0$
3: **for** $t \in [T]$ **do**
4:     $g_t \leftarrow \nabla \mathcal{L}(x_t, \xi_t)$
5:     $m_t \leftarrow \beta_1 m_{t-1} + (1 - \beta_1) g_t \; t \equiv 1 \pmod{\phi}$
6:     $p_t \leftarrow \xi_t$           ▷ Obtain logits from batch
7:     $p_t \leftarrow \texttt{softmax}(p_t)$          ▷ Sample from logits
8:     $\hat{\mathcal{L}}(x_t, \xi_t) \leftarrow p_t$        ▷ Loss, where $p_t$ are labels
9:     $\hat{g}_t \leftarrow \nabla \hat{\mathcal{L}}(x_t, \xi_t)$
10:    $\hat{h}_t \leftarrow |\xi_t| \hat{g}_t \odot \hat{g}_t$
11:    $h_t \leftarrow \beta_2 h_{t-\phi} + (1 - \beta_2) \hat{h}_t$
12:    $h_t \leftarrow h_{t-1}$
13:    $x_{t+1} \leftarrow x_t - \gamma_t \left( \texttt{sign}(m_t) \min\left\{ \frac{|m_t|}{\rho h_t + \varepsilon}, 1 \right\} + \lambda x_t \right)$
14: **end for**
15: **Return:** $x_T$

---

### A.4 `Schedule-Free AdamW`, `Prodigy`

In this section, we outline two more players — `Schedule-Free AdamW` [17] and `Prodigy` [54]. Both of them have a promising advantages and require less hyperparameter tuning which paves the road to parameter-free optimizers.

`Schedule-Free AdamW`. [17] introduced a concept of schedule-free optimizers. Underlying idea behind his `Schedule-Free SGD` and `Schedule-Free AdamW` is to remove the scheduler with iterate averaging. Particularly, schedule-free method uses an interpolation between Polyak-Ruppert averaging [68, 73] and Primal averaging [56] for momentum update instead of usual EMA (`line 4` of Algorithm 12). To avoid undesirable behavior during scalable training the authors also propose internal warmup (see `line 7` of Algorithm 12) which uses the general number of warmup iterations parameter in the code, this gradually increases the learning rate and, at the same time, ensures `Adam`'s bias correction.

An interesting result we observe, `SF-AdamW` shows the best performance with larger number of warmup iterations compared to other methods.

Another key point — training with `SF-AdamW` is sensitive to the choice of beta parameters. Unlike in `AdamW`, these parameters serve different purposes in `SF-AdamW`: as $\beta_1$ acts as an interpolation parameter between two sequences, and $\beta_2$ controls the EMA of the second moment estimate, which relates to $y$ sequence rather than the model parameters $x$ (`line 6` of Algorithm 12). For `Adam` it is common to analyze in theory the case, when $\beta_2 = 1 - 1/T$ [89, 10], i.e., the choice of the "optimal" beta parameters depends on the length of training. Which is also the case for `SF-AdamW`, making it not fully schedule-free. [28] observed this sensitivity to beta parameters, and we go beyond this ablation also.

Importantly, the authors mention that disabling the gradient norm clipping is crucial for schedule-free runs, however, we do not observe this in practice, demonstrating the contrary results.

---

**Algorithm 12** `SF-AdamW`

---

1: **Input:** Initial parameters $x_0$, number of iterations $T$, learning rate $\gamma$, weight decay $\lambda$, $\beta_1$, $\beta_2$, warmup iterations $T_{\text{warmup}}$, $\varepsilon$.
2: **Initialize:** $z_0 \leftarrow x_0$, $v_0 \leftarrow 0$
3: **for** $t \in [T]$ **do**
4:     $y_t \leftarrow (1 - \beta_1)z_t + \beta_1 x_t$
5:     $g_t \leftarrow \nabla \mathcal{L}(y_t, \xi_t)$
6:     $v_t \leftarrow \beta_2 v_{t-1} + (1 - \beta_2) g_t \odot g_t$
7:     $\gamma_t \leftarrow \gamma \sqrt{1 - \beta_2^t} \min\{1, t/T_{\text{warmup}}\}$
8:     $z_{t+1} \leftarrow z_t - \gamma_t \left( g_t / (\sqrt{v_t} + \varepsilon) + \lambda y_t \right)$
9:     $c_{t+1} \leftarrow \frac{\gamma_t^2}{\sum_{i=0}^{t} \gamma_i^2}$
10:     $x_{t+1} \leftarrow (1 - c_{t+1})x_t + c_{t+1} z_{t+1}$
11: **end for**
12: **Return:** $x_T$

---

**Prodigy.** Improving the `D-Adaptation` concept [16], [54] derived an `Adam`-like method, which use an EMA for the learning rate (see `lines 8, 9` of Algorithm 13), approximating the `Adam`'s update of the second moment estimate EMA. The derived update reflects an EMA of $d_t g_t$ sequence rather than $g_t$. Idea of such a method is to come up with a scheme that is able to remove the hand-tuned learning rate via sequence which adapts during training on the fly. In Algorithm 13, $d_t$ is such a sequence that affects both first and second moment estimates, and evolves according to `line 10`.

Crucially, `Prodigy` does not need the learning rate tuning (typically, we initialize $\gamma = 1$), however, it still can be compatible with learning rate schedules, which we verify experimentally at scale. We also show that $d_t$ sequence indeed acts similarly to cosine learning rate scheduler, with usually smaller learning rate at initialization and a bit larger values of it at maximum Moreover, this method scales reliably similar to `AdamW`, making it a promising choice for future development of parameter-free methods.

## A.5 `MARS`

Very recently, [88] introduce `MARS` — a series optimizers, which incorporate modern adaptive methods [50, 9] and approximate second-order methods [24] with variance reduction update update style.

This optimization framework gave a birth to: `MARS-AdamW` — our main baseline which we call simply `MARS`, `MARS-Lion` and `MARS-Shampoo`. We mainly include `MARS-AdamW` in our ablation studies, but report results for other two optimizers.

The authors modified a variance reduction update introducing a sclaing parameter $\eta$, which we call variance reduction scaling in the outlined algorithms and experiments. This parameter controls the scale of gradient correction – see `line 5` of Algorithms 14 to 16.

---

**Algorithm 13** Prodigy

---

1: **Input:** Initial parameters $\boldsymbol{x}_0$, number of iterations $T$, learning rate $\gamma$, weight decay $\lambda$, $\beta_1$, $\beta_2$, $\varepsilon$.
2: **Initialize:** $d_0 \leftarrow 10^{-6}$, $\gamma \leftarrow 1$, $\boldsymbol{m}_0 \leftarrow \boldsymbol{0}$, $\boldsymbol{v}_0 \leftarrow \boldsymbol{0}$, $r_0 \leftarrow 0$, $\boldsymbol{s}_0 \leftarrow \boldsymbol{0}$   ▷ Optional: use scheduler on $\gamma$
3: **for** $t \in [T]$ **do**
4:      $\boldsymbol{g}_t \leftarrow \nabla \mathcal{L}(\boldsymbol{x}_t, \boldsymbol{\xi}_t)$
5:      $\boldsymbol{m}_t \leftarrow \beta_1 \boldsymbol{m}_{t-1} + (1 - \beta_1) d_t \boldsymbol{g}_t$
6:      $\boldsymbol{v}_t \leftarrow \beta_2 \boldsymbol{v}_{t-1} + (1 - \beta_2) d_t^2 \boldsymbol{g}_t \odot \boldsymbol{g}_t$
7:      $\gamma_t \leftarrow \gamma \sqrt{1 - \beta_2^t} / (1 - \beta_1^t)$                           ▷ Optional: use bias correction
8:      $r_t \leftarrow \sqrt{\beta_2} r_{t-1} + (1 - \sqrt{\beta_2}) \gamma_t d_t^2 \langle \boldsymbol{g}_t, \boldsymbol{x}_0 - \boldsymbol{x}_t \rangle$
9:      $\boldsymbol{s}_t \leftarrow \sqrt{\beta_2} \boldsymbol{s}_{t-1} + (1 - \sqrt{\beta_2}) \gamma_t d_t^2 \boldsymbol{g}_t$
10:     $d_{t+1} \leftarrow \max\{d_t, \frac{r_t}{\|\boldsymbol{s}_t\|_1}\}$
11:     $\boldsymbol{x}_{t+1} \leftarrow \boldsymbol{x}_t - \gamma_t d_t \left(\boldsymbol{m}_t / \left(\sqrt{\boldsymbol{v}_t} + d_t \varepsilon\right) + \lambda \boldsymbol{x}_t\right)$
12: **end for**
13: **Return:** $\boldsymbol{x}_T$

---

An important detail, we follow only the approximate scheme of MARS-like optimizers, i.e., we evaluate the gradient $\boldsymbol{g}_t$ in different stochasticity, meaning

$$\boldsymbol{g}_t = \nabla \mathcal{L}\left(\boldsymbol{x}_t, \boldsymbol{\xi}_t\right),$$
$$\boldsymbol{g}_{t-1} = \nabla \mathcal{L}\left(\boldsymbol{x}_{t-1}, \boldsymbol{\xi}_{t-1}\right).$$

Importantly, in the same spirit as for SOAP and Muon, the authors use MARS-like algorithms for layers with two and more dimensions, for 1D layers, embeds, scalar parameters and final the head layer of neural network, this method utilize AdamW. Such a choice allows use MARS in the more fast and still efficient way. Following the practices from their work, we also use MARS only for 2D layers.

MARS (MARS-AdamW). For AdamW-like algorithm, the difference occurs at the computation of $\boldsymbol{m}_t$ and $\boldsymbol{v}_t$, where instead of the gradient, the variance reduction update $\boldsymbol{c}_t$ is used.

---

**Algorithm 14** MARS (MARS-AdamW)

---

1: **Input:** Initial parameters $\boldsymbol{x}_0$, number of iterations $T$, learning rate $\gamma_t$, weight decay $\lambda$, $\beta_1$, $\beta_2$, variance reduction scaling $\eta$, $\varepsilon$.
2: **Initialize:** $\boldsymbol{m}_0 \leftarrow \boldsymbol{0}$, $\boldsymbol{v}_0 \leftarrow \boldsymbol{0}$
3: **for** $t \in [T]$ **do**
4:      $\boldsymbol{g}_t \leftarrow \nabla \mathcal{L}(\boldsymbol{x}_t, \boldsymbol{\xi}_t)$
5:      $\boldsymbol{c}_t \leftarrow \boldsymbol{g}_t + \eta \frac{\beta_1}{1 - \beta_1} \left(\boldsymbol{g}_t - \boldsymbol{g}_{t-1}\right) \|\boldsymbol{c}_t\|_2 > 1$
6:      $\boldsymbol{c}_t \leftarrow \boldsymbol{c}_t / \|\boldsymbol{c}_t\|_2$
7:      $\boldsymbol{m}_t \leftarrow \beta_1 \boldsymbol{m}_{t-1} + (1 - \beta_1) \boldsymbol{c}_t$
8:      $\boldsymbol{v}_t \leftarrow \beta_2 \boldsymbol{v}_{t-1} + (1 - \beta_2) \boldsymbol{c}_t \odot \boldsymbol{c}_t$
9:      $\hat{\boldsymbol{m}}_t \leftarrow \boldsymbol{m}_t / (1 - \beta_1^t)$,    $\hat{\boldsymbol{v}}_t \leftarrow \boldsymbol{v}_t / (1 - \beta_2^t)$
10:     $\boldsymbol{x}_{t+1} = \boldsymbol{x}_t - \gamma_t \left(\frac{\hat{\boldsymbol{m}}_t}{\sqrt{\hat{\boldsymbol{v}}_t} + \varepsilon} + \lambda \boldsymbol{x}_t\right)$
11: **end for**
12: **Return:** $\boldsymbol{x}_T$

---

MARS-Lion. Similarly for Lion-like algorithm, the authors use scaled gradient correction with the current gradient — $\boldsymbol{c}_t$.

MARS-Shampoo. The same holds for MARS-Shampoo. One key comment here, is that to compute SVD of the first moment estimate, the authors also use Newton-Schulz iteration [5, 25]. In our experiments we use 10 iterations of this orthogonalization scheme for MARS-Shampoo.

---

**Algorithm 15** `MARS-Lion`

---

1: **Input:** Initial parameters $\boldsymbol{x}_0$, number of iterations $T$, learning rate $\gamma_t$, weight decay $\lambda$, $\beta_1$, variance reduction scaling $\eta$, $\varepsilon$.
2: **Initialize:** $\boldsymbol{m}_0 \leftarrow \boldsymbol{0}$, $\boldsymbol{v}_0 \leftarrow \boldsymbol{0}$
3: **for** $t \in [T]$ **do**
4:      $\boldsymbol{g}_t \leftarrow \nabla\mathcal{L}(\boldsymbol{x}_t, \boldsymbol{\xi}_t)$
5:      $\boldsymbol{c}_t \leftarrow \boldsymbol{g}_t + \eta\frac{\beta_1}{1-\beta_1}\left(\boldsymbol{g}_t - \boldsymbol{g}_{t-1}\right) \|\boldsymbol{c}_t\|_2 > 1$
6:      $\boldsymbol{c}_t \leftarrow \boldsymbol{c}_t / \|\boldsymbol{c}_t\|_2$
7:      $\boldsymbol{m}_t \leftarrow \beta_1\boldsymbol{m}_{t-1} + (1-\beta_1)\boldsymbol{c}_t$
8:      $\boldsymbol{x}_{t+1} = \boldsymbol{x}_t - \gamma_t\left(\texttt{sign}\left(\boldsymbol{m}_t\right) + \lambda\boldsymbol{x}_t\right)$
9: **end for**
10: **Return:** $\boldsymbol{x}_T$

---

---

**Algorithm 16** `MARS-Shampoo`

---

1: **Input:** Initial parameters $\boldsymbol{x}_0$, number of iterations $T$, learning rate $\gamma_t$, weight decay $\lambda$, $\beta_1$, variance reduction scaling $\eta$, $\varepsilon$.
2: **Initialize:** $\boldsymbol{m}_0 \leftarrow \boldsymbol{0}$, $\boldsymbol{v}_0 \leftarrow \boldsymbol{0}$
3: **for** $t \in [T]$ **do**
4:      $\boldsymbol{g}_t \leftarrow \nabla\mathcal{L}(\boldsymbol{x}_t, \boldsymbol{\xi}_t)$
5:      $\boldsymbol{c}_t \leftarrow \boldsymbol{g}_t + \eta\frac{\beta_1}{1-\beta_1}\left(\boldsymbol{g}_t - \boldsymbol{g}_{t-1}\right)$
6:      $\boldsymbol{m}_t \leftarrow \beta_1\boldsymbol{m}_{t-1} + (1-\beta_1)\boldsymbol{c}_t$
7:      $\boldsymbol{U}_t, \boldsymbol{\Sigma}_t, \boldsymbol{V}_t \leftarrow \texttt{SVD}(\boldsymbol{m}_t)$
8:      $\boldsymbol{x}_{t+1} = \boldsymbol{x}_t - \gamma_t\left(\boldsymbol{U}_t\boldsymbol{V}_t^\top + \lambda\boldsymbol{x}_t\right)$
9: **end for**
10: **Return:** $\boldsymbol{x}_T$

---

## B  Implementation

Our code is based on an extension of nanoGPT[5] and uses PyTorch [63] as well as FlashAttention [14]. We incorporate mixed-precision training [53], i.e., we train in `bfloat16` precision, except for normalization modules and softmax which we train in `float32`. The optimizer states are also stored in `float32`. The majority of the experiments were performed using a cluster of A100-SXM4-80GB GPUs as well as H100-HBM3-80GB. We trained both in a single GPU regime and in DDP [45] (from 2 to 8 GPUs per one run). We estimate that the full cost of all experiments for our project to roughly 30000 GPU hours.

## C  Model & Data

**Architecture details.** In our project, we use Llama-like family of models [48]. We implement the the popular in the community decoder-only transformer with SwiGLU activation functions [77], RoPE embeddings [81], RMSNorm [91]. The vocabulary is based on the GPT2 [71] tokenizer [6] and contains 50304 tokens.

The number of parameters in our models is fully configurable, and we present the exact configurations used in our experiment in Table 2.

**Dataset.** Our main findings are obtained on the subset of FineWeb [64] with $100\mathbf{B}$ tokens [7], cleaned and deduplicated corpus for LLM pretraining, which we split into train and validation sequences. During training, we evaluate the models with a fixed set of 32 batches of our chosen sequence length (512 for almost all experiments, the same context length as training) to establish the validation loss curves. At the end of training, we compute the full validation loss and perplexity (this loss is reported

---

[5]https://github.com/karpathy/nanoGPT
[6]https://github.com/openai/tiktoken
[7]https://huggingface.co/datasets/HuggingFaceFW/fineweb

Table 2: Hyperparameters for our Llama-like models.

| # Parameters | 124M | 210M | 600M |
|---|---|---|---|
| Hidden size | 768 | 768 | |
| # Attention heads | 12 | 12 | |
| # Layers | 12 | 24 | |
| Init std | 0.02 | 0.02 | 0.02 |
| Use bias | no | no | no |
| RMSNorm epsilon | 0.00001 | 0.00001 | 0.00001 |
| Positional encoding | RoPE | RoPE | RoPE |

as Final Validation Loss in the figures). We also performed our initial results on the subset of OpenWebText2 dataset [21].

# D  Hyperparameter tuning

**How do we tune hyperparameters?**    We perform systematic hyperparameter tuning for all algorithms, starting with smaller models (124M, 210M) and extrapolating to larger ones. Our tuning process focused on two primary settings: **Small batch setting** (32 batch size) and **Large batch setting** (256 batch size). For both settings, we use a sequence length of 512 tokens, resulting in 16k and 130k tokens per batch, respectively. If the batch cannot fit into memory, we use gradient accumulation steps, while maintaining the effective batch size.

We also include ablations on even larger batch size for 124M models, where we train on 512 batch size (260k tokens correspondingly). And larger, 583M models, we train on 3936 batch size, preserving the basic sequence length of 512, i.e., 4M tokens.

We first run multiple experiments, greed searching hyperparameters, on near Chinchilla optimal length of training using *cosine learning rate scheduler* (except for `SF-AdamW`):

• for 124M models we tune at 2.1B tokens for both small (32) and large (256) batch size setting,

• for 210M models we tune at 4.2B tokens for our large batch size setting,

• for 583M models we also consider a setting with and without z-loss.

We present the configurations for different training horizons in Tables 3 and 5.

Table 3: Lengths of training for **Small batch settings** ($32 \times 512$).

| # Parameters | Tokens (Iterations) | | | | | | Ch. Tokens |
|---|---|---|---|---|---|---|---|
| 124M | 1B (64k) | 2.1B (128k) | 4.2B (256k) | 6.3B (384k) | 8.4B (512k) | 16.8B (1024k) | 2.5B |
| 210M | 1B (64k) | 2.1B (128k) | 4.2B (256k) | 6.3B (384k) | 8.4B (512k) | 16.8B (1024k) | 4.2B |

Table 4: Lengths of training for **Large batch settings** ($256 \times 512$).

| # Parameters | Tokens (Iterations) | | | | | | Chinchilla Tokens |
|---|---|---|---|---|---|---|---|
| 124M | 1B (8k) | 2.1B (16k) | 4.2B (32k) | 6.3B (48k) | 8.4B (64k) | 16.8B (128k) | 2.5B |
| 210M | 1B (8k) | 2.1B (16k) | 4.2B (32k) | 6.3B (48k) | 8.4B (64k) | 16.8B (128k) | 4.2B |

Important to note, for larger models, we mostly kept the best hyperparameters found for the 124M model and re-tuned the learning rate and gradient clipping. We summarize this process in Appendices D.1 to D.3.

Additionally, when we report an of one particular hyperparameters, we mean that corresponding algorithm has already been tuned and, thus, we show only how one particular hyperparameter affects the overall performance.

**Hyperparameters used in our WSD scheduler experiments.**    Once we found the best setting for each method using cosine learning rate scheduler, we are ready to obtain the optimal performance of our method with WSD scheduler [27]. Here we follow the rule of thumb from [28]:

Table 5: Lengths of training for **X-Large batch settings** ($1984 \times 512$).

| # Parameters | Tokens (Iterations) | | | Chinchilla Tokens |
|:---:|:---:|:---:|:---:|:---:|
| 720M | 8B (8k) | 16B (16k) | 48B (48k) | 14.4B |

- use half the optimal learning rate for the cosine scheduler,

- use $20\%$ of iterations for cooldown phase,

- use $(1 - \sqrt{x})$ decay shape for the cooldown phase,

the only difference is that we do not employ stochastic weight averaging [29].

Therefore, we maintain most hyperparameters across optimizers, only re-tuning the learning rate. For methods like `Muon` and `MARS`, we reduce both `AdamW`'s learning rate and the learning rate for non-1D parameters. This approach ensures a fair comparison while accounting for the unique properties of each optimizer.

Indeed, this rule of thumb works better in our setting also, e.g., see the comparison between linear decay shape and $(1 - \sqrt{x})$ in

We report a series of comparisons between cosine learning rate scheduler and WSD in

**Hyperparameters used in `z-loss` experiments.**

**D.1  124M parameters model**

Table 6: `AdamW` hyperparameter tuning for our 124M parameter large language models. Bold hyperparameters are the best.

| Hyperparameter | Small batch setting | Large batch setting |
|:---:|:---:|:---:|
| Learning rate | $0.0001, \mathbf{0.0005}, 0.0008, 0.001, 0.002$ | $0.0001, 0.0003, 0.0005, \mathbf{0.001}, 0.002$ |
| Batch size | 32 | 256 |
| Sequence length | 512 | 512 |
| Number of warmup steps | $\mathbf{3000}, 5000, 8000$ | $500, 1000, \mathbf{2000}, 3000, 8000, 32000$ |
| Weight decay | 0.1 | no, $0.1, \mathbf{0.5}, 0.7$ |
| Learning rate decay scheduler | WSD, **cosine** | WSD, **cosine** |
| Gradient clipping | no, $\mathbf{0.5}, 1, 1.5$ | no, $0.5, 1$ |
| AdamW $\beta_1$ | $0.5, \mathbf{0.8}, 0.9$ | $\mathbf{0.8}, 0.9$ |
| AdamW $\beta_2$ | $0.95, \mathbf{0.999}$ | $0.95, 0.99, \mathbf{0.999}, 0.9999$ |

Table 7: `ADOPT` hyperparameter tuning for our 124M parameter large language models. Bold hyperparameters are the best.

| Hyperparameter | Small batch setting | Large batch setting |
|:---:|:---:|:---:|
| Learning rate | 0.001 | $0.0001, 0.0003, 0.0005, \mathbf{0.001}, 0.002$ |
| Batch size | 32 | 256 |
| Sequence length | 512 | 512 |
| Number of warmup steps | $\mathbf{3000}, 8000$ | $\mathbf{2000}, 8000, 32000$ |
| Weight decay | 0.1 | no $\mathbf{0.1}, 0.5$ |
| Learning rate decay scheduler | WSD, **cosine** | WSD, **cosine** |
| Gradient clipping | 0.5 | no, $0.5, 1$ |
| ADOPT $\beta_1$ | 0.9 | $0.8, \mathbf{0.9}$ |
| ADOPT $\beta_2$ | $\mathbf{0.999}, 0.9999$ | $0.5, \mathbf{0.999}, 0.9999$ |

Table 8: `AdEMAMix` hyperparameter tuning for our 124M parameter large language models. Bold hyperparameters are the best.

| Hyperparameter | Small batch setting | Large batch setting |
|---|---|---|
| Learning rate | $0.0001, \mathbf{0.0005}, 0.0008, 0.001, 0.002$ | $0.0001, 0.0003, 0.0005, \mathbf{0.001}, 0.002$ |
| Batch size | 32 | 256 |
| Sequence length | 512 | 512 |
| Number of warmup steps | $\mathbf{3000}, 8000$ | $\mathbf{2000}, 8000, 32000$ |
| Weight decay | 0.1 | no, $0.1, \mathbf{0.5}, 0.7$ |
| Learning rate decay scheduler | WSD, **cosine** | WSD, **cosine** |
| Gradient clipping | no, $\mathbf{0.5}, 1, 1.5$ | no, $\mathbf{0.5}, 1$ |
| `AdEMAMix` $\beta_1$ | $0.5, \mathbf{0.8}, 0.9$ | $0.8, \mathbf{0.9}$ |
| `AdEMAMix` $\beta_2$ | $0.999$ | $\mathbf{0.999}, 0.9999$ |
| `AdEMAMix` $\beta_3$ | $0.999, \mathbf{0.9999}, 0.99995$ | $\mathbf{0.999}, 0.9999$ |
| `AdEMAMix` $\alpha$ | $5, \mathbf{8}, 12$ | 8 |

Table 9: `Lion` hyperparameter tuning for our 124M parameter large language models. Bold hyperparameters are the best.

| Hyperparameter | Small batch setting | Large batch setting |
|---|---|---|
| Learning rate | $0.00005, \mathbf{0.0001}, 0.0005, 0.001$ | $0.0001, 0.0005, \mathbf{0.001}, 0.002$ |
| Batch size | 32 | 256 |
| Sequence length | 512 | 512 |
| Number of warmup steps | 3000 | $\mathbf{2000}, 8000, 32000$ |
| Weight decay | no, $0.1, 0.2, \mathbf{0.5}$ | no, $0.1, \mathbf{0.5}, 0.7$ |
| Learning rate decay scheduler | WSD, **cosine** | WSD, **cosine** |
| Gradient clipping | 0.5 | no, $\mathbf{0.5}, 1$ |
| `Lion` $\beta_1$ | $0.7, \mathbf{0.9}, 0.99$ | $0.5, \mathbf{0.9}$ |
| `Lion` $\beta_2$ | $0.9, \mathbf{0.99}, 0.999$ | $\mathbf{0.99}, 0.999$ |

## D.2  210M **parameters model**

## D.3  600M **parameters model**

# E   Additional results

## E.1   **Benchmarking:** 124M

In this section, we provide complete results for the benchmarking part presented in Section 4.1. We cover both the **large batch setting** and the **small batch setting**, reporting the full curves with validation loss dynamics across different training durations.

Given the quite a lot number of methods under consideration, we divide them into two groups: those that outperform `AdamW` and those that underperform relative to `AdamW`. We use `AdamW` loss curves as the reference point in both figures. We summarize our findings for the small batch size of 32 in Figure 12. And for the large batch size of 256 in Figures 13 and 14.

## E.2   **Ablations for** 124M **model**

**Fail of** `Sophia`**.**

**Clipping &** `SF-AdamW`**.**

**Betas sensitivity.**

**Warmup ablation.**   In this section we detaily describe the main part in Section 4.1.

We study the impact of batch size on the final validation loss obtained. For all methods, we sweep over warmup lengths of $\{1.56\%, 6.25\%, 25\%\}$ of the total training duration to examine each method's

Table 10: `Signum` hyperparameter tuning for our 124M parameter large language models. Bold hyperparameters are the best.

| Hyperparameter | Small batch setting | Large batch setting |
|---|---|---|
| Learning rate | $0.0003, 0.0005, \mathbf{0.001}$ | $0.0001, 0.00030.0005, 0.0003, \mathbf{0.001}, 0.002$ |
| Batch size | 32 | 256 |
| Sequence length | 512 | 512 |
| Number of warmup steps | $2000, \mathbf{3000}$ | $\mathbf{2000}, 8000, 32000$ |
| Weight decay | $\text{no}, 0, \mathbf{0.1}, 0.5$ | $\text{no}, 0, \mathbf{0.1}, 0.5, 0.7$ |
| Learning rate decay scheduler | WSD, **cosine** | WSD, **cosine** |
| Gradient clipping | $\text{no}, \mathbf{0.5}, 1$ | $\text{no}, \mathbf{0.5}, 1$ |
| Momentum | $\text{no}, 0.9, \mathbf{0.95}$ | $\text{no}, 0.9, \mathbf{0.95}, 0.99$ |
| Nesterov momentum | no, **yes** | no, **yes** |

Table 11: `Muon` hyperparameter tuning for our 124M parameter large language models. Bold hyperparameters are the best.

| Hyperparameter | Small batch setting | Large batch setting |
|---|---|---|
| Learning rate `AdamW` | $0.0001, 0.0003, 0.0005, \mathbf{0.001}, 0.002$ | $0.0001, 0.0003, 0.0005, \mathbf{0.001}, 0.002$ |
| Learning rate `Muon` | $0.001, \mathbf{0.01}, 0.02$ | $0.001, \mathbf{0.01}, 0.02$ |
| Batch size | 32 | 256 |
| Sequence length | 512 | 512 |
| Number of warmup steps | $\mathbf{3000}, 8000$ | $\mathbf{2000}, 8000, 32000$ |
| Weight decay | $\text{no}, \mathbf{0.1}, 0.5$ | $\text{no}, \mathbf{0.1}, 0.5$ |
| Learning rate decay scheduler | **WSD**, cosine | **WSD**, cosine |
| Gradient clipping | $\text{no}, \mathbf{0.5}$ | $\text{no}, \mathbf{0.5}, 1.0$ |
| Momentum `Muon` | $0.9, 0.95, \mathbf{0.99}$ | $\mathbf{0.95}, 0.99$ |
| Optimizer for 1D layers | `AdamW` | `AdamW` |
| Optimizer for 1D layers, $\beta_1$ | $0.8, 0.9$ | $\mathbf{0.8}, 0.9$ |
| Optimizer for 1D layers, $\beta_2$ | $0.99, \mathbf{0.999}, 0.9999$ | $0.99, 0.999, 0.9999$ |
| Newton-Schulz a | 3.4445 | 3.4445 |
| Newton-Schultz b | $-4.7750$ | $-4.7750$ |
| Newton-Schultz c | 2.0315 | 2.0315 |
| Nesterov momentum | no, **yes** | no, **yes** |

sensitivity to warmup. For `AdamW`, we extend this sweep to $\{1.56\%, 5\%, 6.25\%, 10\%, 25\%\}$. We specifically consider the $1.56\%$ and $6.25\%$ percentages because the former represents a typical number of warmup steps (2000) for models of our scale, while the latter ($6.25\%$ of 128000 steps) aligns with the warmup strategy used in Llama [48].

Contrary to the insights from [92], we observe that $25\%$ of the Cinchilla optimal duration is far from being the best batch size for pretraining. We emphasize that their results were obtained for 85M models and then extrapolated to larger scales. However, in our setting, we found the basic 2000 steps a more suitable option for warmup. $25\%$ of Chinchilla optimal length of training, for our 124M model is 620M tokens.

We provide the warmup sweep for `AdamW` in Figure 19

In addition to validating the results from [92], we report the sensitivity of different optimizers to the number of warmup steps by conducting a sweep over the aforementioned percentages. A summary of this experiment — Figure 6.

`Muon` **Newton-Schulz iterations.**

**Weight decay ablation.**

**Learning rate sensitivity.** In this part of the work, we meticulously replicate the learning rate sweep process and present comprehensive results. Consistent with our experimental setup, we aim to determine the true impact of the learning rate and its transferability to longer training horizons.

Table 12: `SOAP` hyperparameter tuning for our 124M parameter large language models. Bold hyperparameters are the best.

| Hyperparameter | Small batch setting | Large batch setting |
|---|---|---|
| Learning rate | $0.005, \mathbf{0.001}$ | $0.0001, 0.0003, 0.0005, \mathbf{0.001}, 0.002$ |
| Batch size | 32 | 256 |
| Sequence length | 512 | 512 |
| Number of warmup steps | $\mathbf{3000}, 8000$ | $\mathbf{2000}, 4000, 8000, 12000, 16000, 32000$ |
| Weight decay | 0.1 | no, $\mathbf{0.1}, 0.5$ |
| Learning rate decay scheduler | WSD, **cosine** | WSD, **cosine** |
| Gradient clipping | 0.5 | no, $\mathbf{0.5}, 1$ |
| Preconditioner dimension | 10000 | 10000 |
| Preconditioning frequency | $1, 5, \mathbf{10}$ | $1, 5, \mathbf{10}$ |
| `SOAP` $\beta_1$ | $0.8, \mathbf{0.9}$ | $0.8, \mathbf{0.9}, 0.95$ |
| `SOAP` $\beta_2$ | $0.95, 0.99, \mathbf{0.999}, 0.9999$ | $0.95, 0.99, \mathbf{0.999}, 0.9999$ |

Table 13: `Sophia` hyperparameter tuning for our 124M parameter large language models. Bold hyperparameters are the best.

| Hyperparameter | Small batch setting | Large batch setting |
|---|---|---|
| Learning rate | $0.0001, \mathbf{0.0003}, 0.0005, 0.001, 0.002$ | $0.0001, 0.0003, 0.0005, \mathbf{0.001}, 0.002, 0.01$ |
| Batch size | 32 | 256 |
| Sequence length | 512 | 512 |
| Number of warmup steps | $\mathbf{2000}, 3000$ | $\mathbf{2000}, 8000, 32000$ |
| Weight decay | 0.1 | no, $\mathbf{0.1}, 0.5$ |
| Learning rate decay scheduler | WSD, **cosine** | WSD, **cosine** |
| Gradient clipping | 0.5 | no, $\mathbf{0.5}, 1$ |
| Estimator | Gauss-Newton-Bartlett | Gauss-Newton-Bartlett |
| Estimator frequency | 10 | 10 |
| `Sophia` $\beta_1$ | 0.9 | $0.8, \mathbf{0.9}$ |
| `Sophia` $\beta_2$ | $0.95, \mathbf{0.999}, 0.9999, 0.99999$ | $0.95, \mathbf{0.999}, 0.9999, 0.99999$ |
| `Sophia` $\rho$ | $0, 0.03, \mathbf{0.04}$ | $0, 0.03, \mathbf{0.04}$ |

For each optimizer (except `Prodigy`), we vary only the learning rate while maintaining the best hyperparameter settings obtained during our initial tuning (see Appendices D and D.1) on 2.1B tokens for the 124M parameter model. We present the results of the learning rate sweep in Figure 24.

**Cosine vs WSD.** We present our results for two batch size settings: 32 and 256. At first, our initial results in small batch setting on the OpenWebText2 (OWT2) dataset, we present in Figure 25.

We report the final validation loss on the FineWeb dataset for 124M model trained on the batch size of 256. We use our tuned with cosine scheduler methods. For WSD, we follow the rule of thumb from [28]: 20% of steps for the cooldown, $1 - \sqrt{x}$ decay shape, and the learning rate is half the optimal for cosine, i.e., 0.0005 if we have the best learning rate 0.001 for the method. Additionally, we point out that we do not include stochastic weight averaging in the comparison, which might potentially enhance the performance of optimizers with WSD.

**Gradient norm patterns.**

**E.3   Benchmarking: 210M**

**E.4   Ablations for 210M model**

**E.5   Wall-clock performance of optimizers across models of different scale**

**E.6   Extension to MoEs.**

Table 14: `Schedule-Free AdamW` hyperparameter tuning for our $124\text{M}$ parameter large language models. Bold hyperparameters are the best.

| Hyperparameter | Small batch setting | Large batch setting |
|---|---|---|
| Learning rate | $0.0001, 0.0003, 0.0005, \mathbf{0.001}, 0.005$ | $0.0001, 0.0003, 0.0005, \mathbf{0.001}, 0.002, 0.005$ |
| Batch size | 32 | 256 |
| Sequence length | 512 | 512 |
| Number of warmup steps | $\mathbf{3000}, 8000$ | $2000, 4000, \mathbf{8000}, 12000, 16000, 32000$ |
| Weight decay | no, $0.05, \mathbf{0.1}, 0.5$ | no, $0.05, \mathbf{0.1}, 0.5$ |
| Learning rate decay scheduler | no | no |
| Gradient clipping | no, $\mathbf{0.5}$ | no, $\mathbf{0.5}, 1$ |
| `Schedule-Free AdamW` $\beta_1$ | $\mathbf{0.9}, 0.95, 0.98$ | $\mathbf{0.9}, 0.95, 0.98$ |
| `Schedule-Free AdamW` $\beta_2$ | $0.95, 0.99, 0.999, \mathbf{0.9999}, 0.99999$ | $0.95, 0.99, 0.999, \mathbf{0.9999}, 0.99999$ |

Table 15: `Prodigy` hyperparameter tuning for our $124\text{M}$ parameter large language models. Bold hyperparameters are the best.

| Hyperparameter | Small batch setting | Large batch setting |
|---|---|---|
| Learning rate | $0.5, \mathbf{1}$ | $0.5, \mathbf{1}, 2, 10, 100$ |
| Batch size | 32 | 256 |
| Sequence length | 512 | 512 |
| Number of warmup steps | $\mathbf{3000}, 8000$ | $\mathbf{2000}, 4000, 8000, 12000, 16000, 32000$ |
| Weight decay | no, $\mathbf{0.1}, 0.5$ | no, $\mathbf{0.1}, 0.5$ |
| Learning rate decay scheduler | no, WSD, **cosine** | no, WSD, **cosine** |
| Gradient clipping | no, $\mathbf{0.5}, 1$ | no, $\mathbf{0.5}, 1$ |
| `Prodigy` $\beta_1$ | $0.9$ | $0.8, \mathbf{0.9}$ |
| `Prodigy` $\beta_2$ | $0.99, \mathbf{0.999}, 0.9999$ | $\mathbf{0.999}, 0.9999$ |
| `Prodigy` bias correction | no, **yes** | no, **yes** |

Table 16: `MARS (MARS-AdamW)` hyperparameter tuning for our $124\text{M}$ parameter large language models. Bold hyperparameters are the best.

| Hyperparameter | Small batch setting | Large batch setting |
|---|---|---|
| Learning rate `AdamW` | $0.0001, 0.0005, \mathbf{0.001}, 0.003$ | $0.0001, 0.0005, \mathbf{0.001}, 0.003$ |
| Learning rate `MARS` | $0.001, \mathbf{0.003}$ | $0.001, \mathbf{0.003}$ |
| Batch size | 32 | 256 |
| Sequence length | 512 | 512 |
| Number of warmup steps | $2000, \mathbf{3000}$ | $\mathbf{2000}, 8000, 32000$ |
| Weight decay `MARS` | no, $\mathbf{0.1}$ | no, $\mathbf{0.1}, 0.5$ |
| Weight decay for 1D layers | $0.1$ | $0.1$ |
| Learning rate decay scheduler | WSD, **cosine** | WSD, **cosine** |
| Gradient clipping | $0.5$ | $0.5$ |
| Optimizer for 1D layers | `AdamW` | `AdamW` |
| Optimizer for 1D layers $\beta_1$ | $\mathbf{0.8}, 0.9$ | $\mathbf{0.8}, 0.9, 0.95$ |
| Optimizer for 1D layers $\beta_2$ | $0.95, 0.99, \mathbf{0.999}$ | $0.95, 0.99, \mathbf{0.999}$ |
| `MARS` $\beta_1$ | $0.9, \mathbf{0.95}$ | $0.9, \mathbf{0.95}$ |
| `MARS` $\beta_2$ | $0.95, \mathbf{0.99}$ | $0.95, \mathbf{0.99}$ |
| VR scaling factor $\eta$ | $0.024, \mathbf{0.025}$ | $0.024, \mathbf{0.025}$ |

Table 17: `MARS-Lion` hyperparameter tuning for our 124M parameter large language models. Bold hyperparameters are the best.

| Hyperparameter | Small batch setting | Large batch setting |
|---|---|---|
| Learning rate `Lion` | **0.0001**, 0.0005, 0.001, 0.003 | **0.0001**, 0.0005, 0.001, 0.003 |
| Learning rate `MARS` | **0.0001**, 0.001, 0.003 | **0.0001**, 0.001, 0.003 |
| Batch size | 32 | 256 |
| Sequence length | 512 | 512 |
| Number of warmup steps | 2000, **3000** | **2000**, 8000, 32000 |
| Weight decay `MARS` | no, **0.1** | no, **0.1**, 0.5 |
| Weight decay for 1D layers | 0.1 | 0.1 |
| Learning rate decay scheduler | **WSD**, cosine | **WSD**, cosine |
| Gradient clipping | 0.5 | 0.5 |
| Optimizer for 1D layers | `Lion` | `Lion` |
| Optimizer for 1D layers $\beta_1$ | 0.8, **0.9** | 0.8, **0.9**, 0.95 |
| Optimizer for 1D layers $\beta_2$ | 0.95, 0.99, **0.999** | 0.95, 0.99, **0.999** |
| `MARS` $\beta_1$ | 0.9, **0.95** | 0.9, **0.95** |
| `MARS` $\beta_2$ | 0.95, **0.99** | 0.95, **0.99** |
| VR scaling factor $\eta$ | 0.024, **0.025** | 0.024, **0.025** |

Table 18: `MARS-Shampoo` hyperparameter tuning for our 124M parameter large language models. Bold hyperparameters are the best.

| Hyperparameter | Small batch setting | Large batch setting |
|---|---|---|
| Learning rate `Shampoo` | 0.0001, 0.0005, **0.001**, 0.003 | 0.0001, 0.0005, **0.001**, 0.003 |
| Learning rate `MARS` | 0.001, **0.003** | 0.001, **0.003** |
| Batch size | 32 | 256 |
| Sequence length | 512 | 512 |
| Number of warmup steps | 2000, **3000** | **2000**, 8000, 32000 |
| Weight decay `MARS` | no, **0.1** | no, **0.1**, 0.5 |
| Weight decay for 1D layers | 0.1 | 0.1 |
| Learning rate decay scheduler | WSD, **cosine** | WSD, **cosine** |
| Gradient clipping | 0.5 | 0.5 |
| Optimizer for 1D layers | `Shampoo` | `Shampoo` |
| Optimizer for 1D layers $\beta_1$ | 0.8, **0.9** | 0.8, **0.9**, 0.95 |
| Optimizer for 1D layers $\beta_2$ | 0.95, 0.99, **0.999** | 0.95, 0.99, **0.999** |
| `MARS` $\beta_1$ | 0.9, **0.95** | 0.9, **0.95** |
| `MARS` $\beta_2$ | 0.95, **0.99** | 0.95, **0.99** |
| VR scaling factor $\eta$ | 0.024, **0.025** | 0.024, **0.025** |

Table 19: Hyperparameters for our Llama-like models for the wall-clock experiments.

| # Parameters | 30M | 52M | 80M | 124M | 150M | 210M | 360M | 720M | 1B |
|---|---|---|---|---|---|---|---|---|---|
| Hidden size | 384 | 512 | 768 | 768 | 768 | 768 | 1024 | 2048 | 1792 |
| # Attention heads | 6 | 8 | 6 | 12 | 12 | 12 | 16 | 16 | 14 |
| # Layers | 8 | 8 | 6 | 12 | 16 | 24 | 24 | 12 | 24 |
| Init std | 0.02 | 0.02 | 0.02 | 0.02 | 0.02 | 0.02 | 0.02 | 0.02 | 0.02 |
| Use bias | no | no | no | no | no | no | no | no | no |
| RMSNorm epsilon | 0.00001 | 0.00001 | 0.00001 | 0.00001 | 0.00001 | 0.00001 | 0.00001 | 0.00001 | 0.00001 |
| Positional encoding | RoPE | RoPE | RoPE | RoPE | RoPE | RoPE | RoPE | RoPE | RoPE |

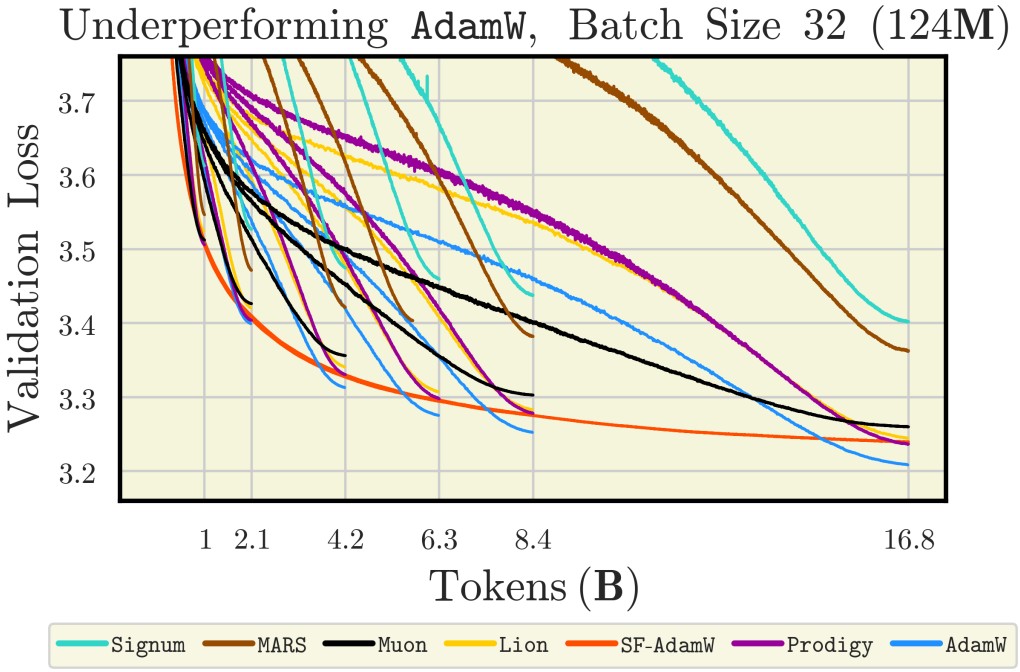

Figure 11: **Ranking of optimizers in the small-batch setting.** In the small-batch setting `AdamW` outperforms most of the optimizaers we study.

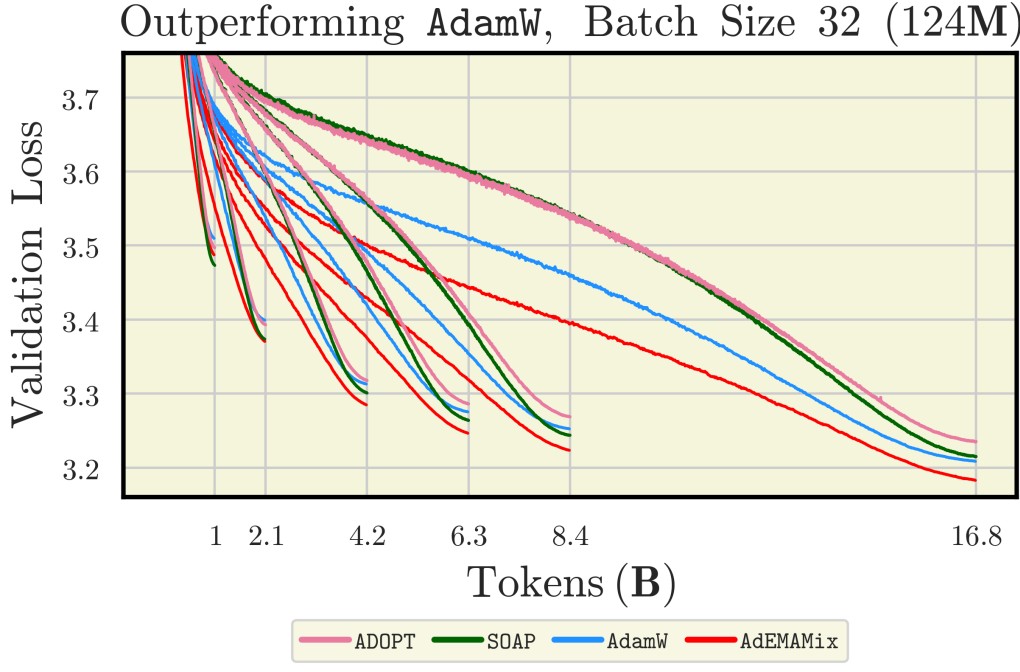

Figure 12: **Ranking of optimizers in the small-batch setting.** Here only `AdEMAMix`, `SOAP` and `ADOPT` show a remarkable performance compared to `AdamW`.

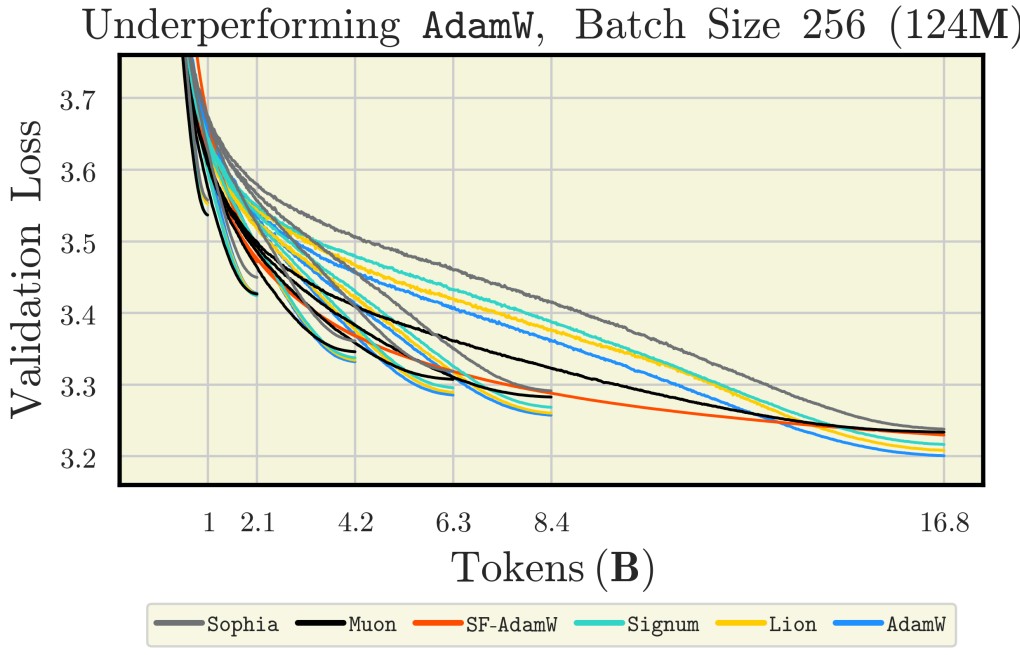

Figure 13: **Ranking of optimizers in the large-batch setting.**

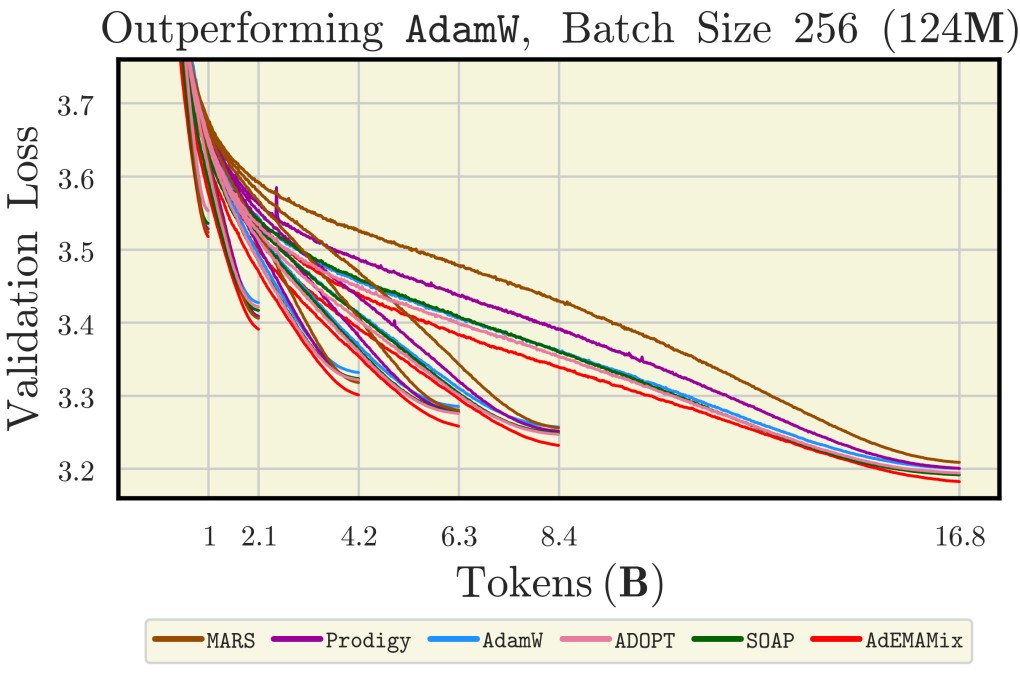

Figure 14: **Ranking of optimizers in the large-batch setting.**

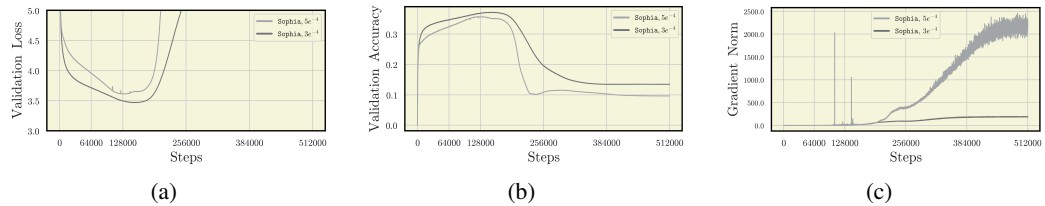

Figure 15: `Sophia` **diverges in the small-batch setting even with sufficiently small learning rate.**

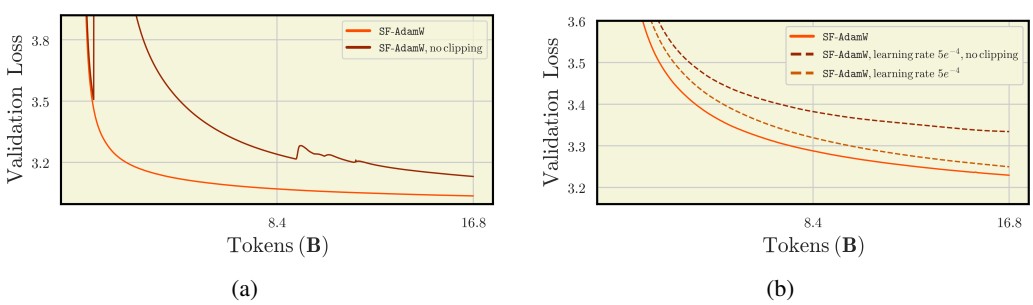

Figure 16: **Clipping is significant for** `Schedule-Free`.

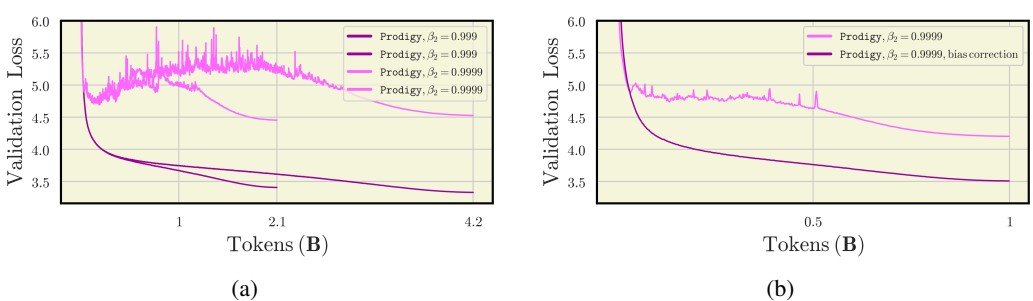

Figure 17: `Prodigy` **is sensitive to beta parameters in the small-batch setting**.

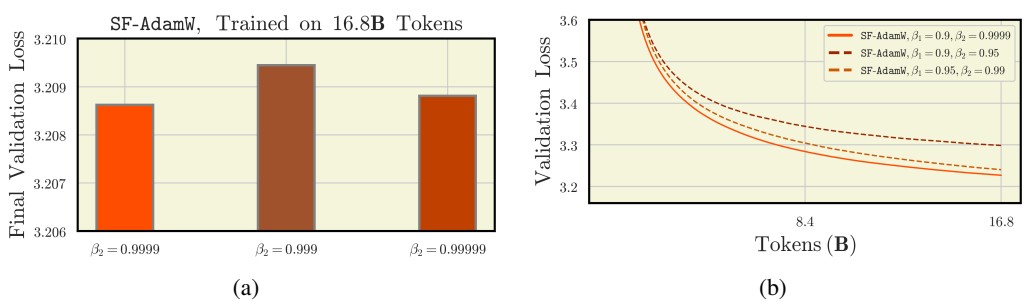

Figure 18: **Impact of beta parameters on** `Schedule-Free`.

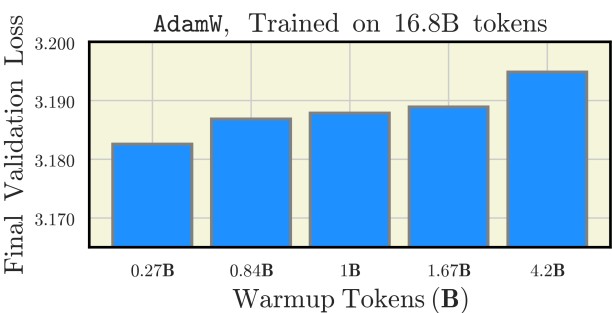

Figure 19: **Warmup sweep for** `AdamW`**.** We observe that the smaller yet reasonable warmup value is the best, however, this is not true for other methods like `Signum` and `SF-AdamW` (see **??**).

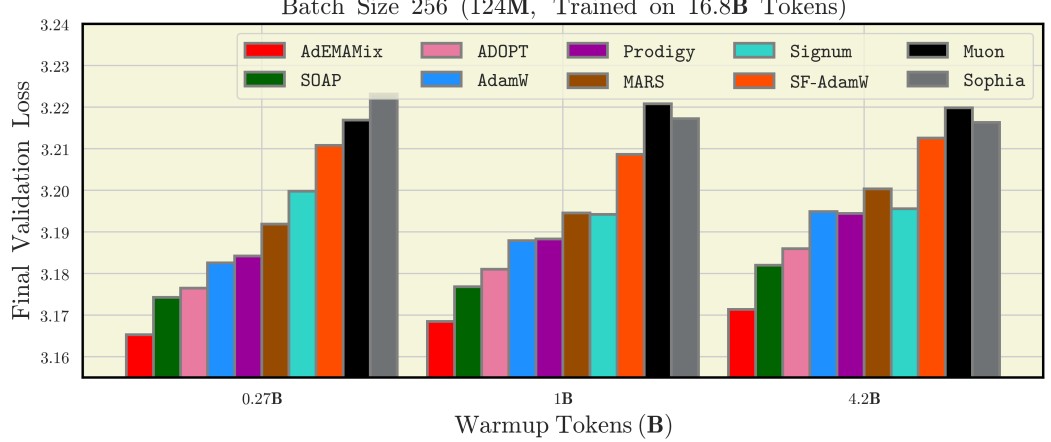

Figure 20: **Warmup ablation.** We report the final validation loss on the FineWeb dataset for 124**M** model trained on the batch size of 256. We sweep over the batch sizes of $\{1.56\%, 6.25\%, 25\%\}$ of the length of training, which corresponds to $\{2000, 8000, 32000\}$**k** iterations, respectively.

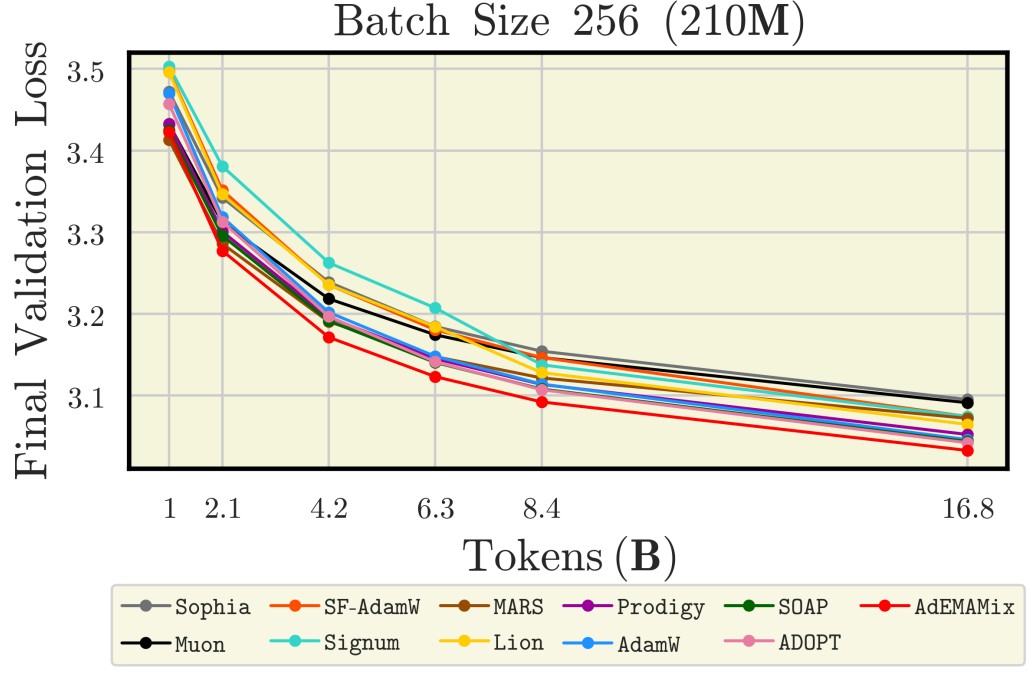

Figure 21: **Ranking of optimizers in the large-batch setting.**

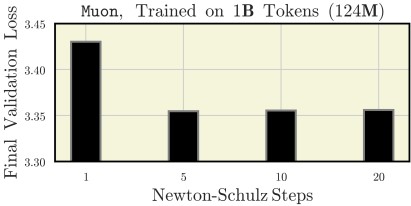

Figure 22: Muon**'s dependence on the number of Newton-Schulz iterations.**

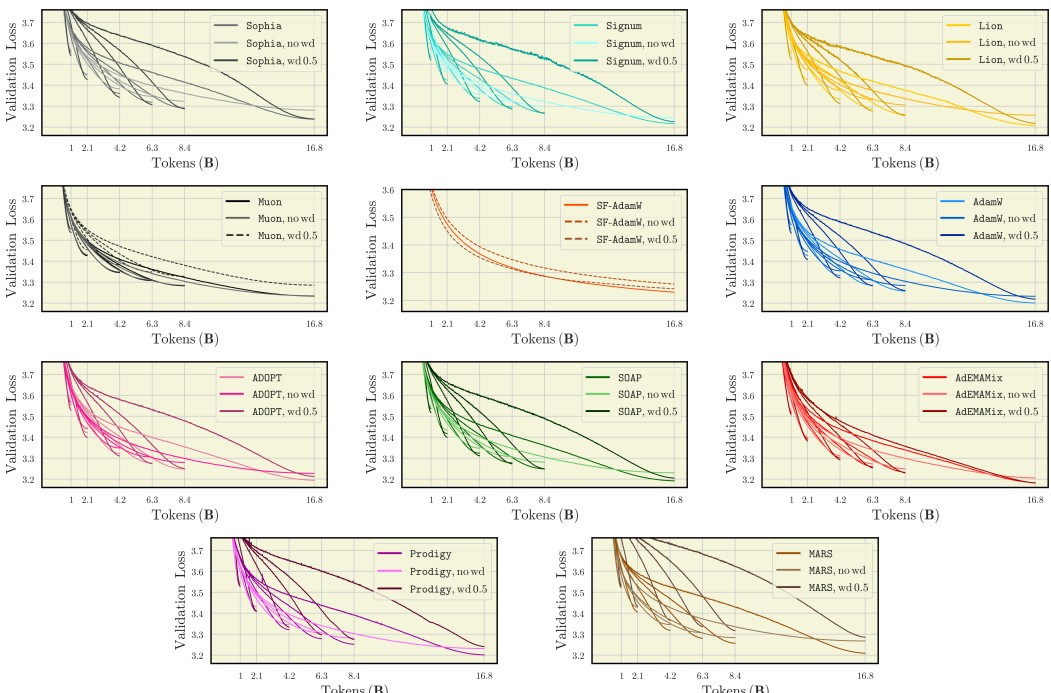

Figure 23: **Larger weight decay achieves significantly better results when training on fewer tokens.** We report the final validation loss on the FineWeb dataset for $124M$ model trained on the batch size of $256$. We observe that the mmajority of runs with the large weight decay of $0.5$ consistently outperform the same optimizer with weight decay of $0.1$ for all training durations except for the last one. Notably, `Signum` and `Lion` with large weight decay perform even better than `AdamW` with the same learning rate. We also consider a setting without weight decay. We observe that this is suboptimal for most of other optimizers, while the typical weight decay of $0.1$ remains the best for large training durations. An interesting thing we observe for optimizers that train one dimensional and two dimensional parameters in a different way — `Muon`, `MARS`. Indeed, the corresponding runs with the weight decay of $0.5$ are always worse than then $0.1$ baseline and, in some cases, even worse than runs without weight decay. For `Muon`, we connect this effect to its algorithmic design, where weight decay is not used to optimize matrix parameters (see Algorithm 7). For `MARS`, we only vary the weight decay that corresponds to matrix parameters, while keeping $0.1$ for all scalar, one dimensional and final layer parameters. In this case, we conclude that the gap between large and small weight decay values narrows significantly faster.

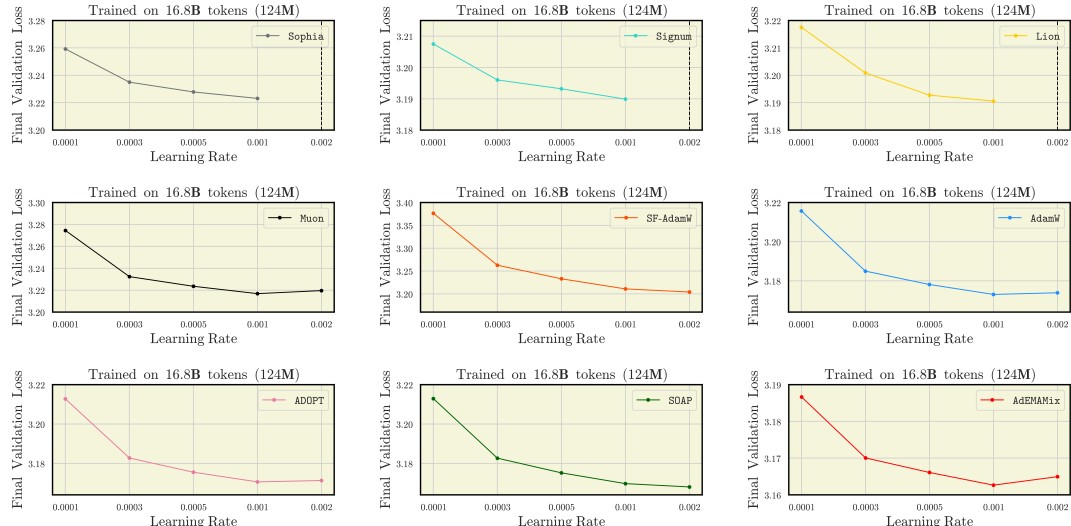

Figure 24: **Learning rate sensitivity.** We report the final validation loss on the FineWeb dataset for $124\mathbf{M}$ model trained on the batch size of $256$. In the current setting, only `SOAP` and `SF-AdamW` reach the better performance with the large leraning rate of $0.02$. On the other hand, `Sophia` and all sign-based methods (`Signum` and `Lion`) diverge with this value of the learning rate.

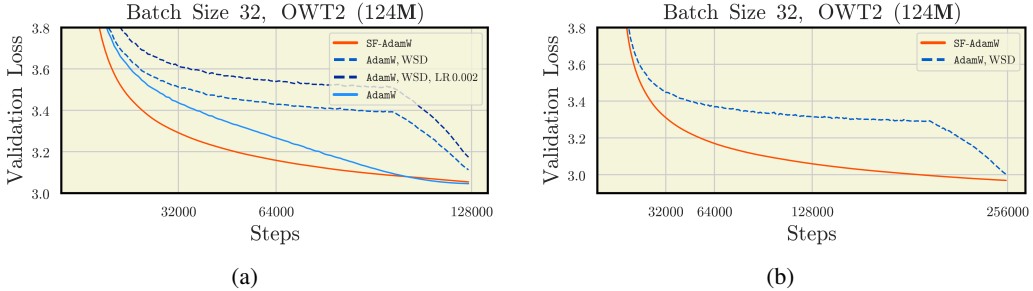

|        (a)        |        (b)        |

Figure 25: **WSD scheduler underperforms both `AdamW` with cosine scheduler and `SF-AdamW`.** Once the learning rate and beta parameters of `SF-AdamW` and `AdamW` are properly tuned, we observe a surprisingly large gap in performance between WSD scheduler and its competitors. Figure **(b)** suggests that this gap may potentially diminish with extended training. To investigate this further, we conduct a scalable comparison between tuned WSD and cosine baselines across longer training horizons.

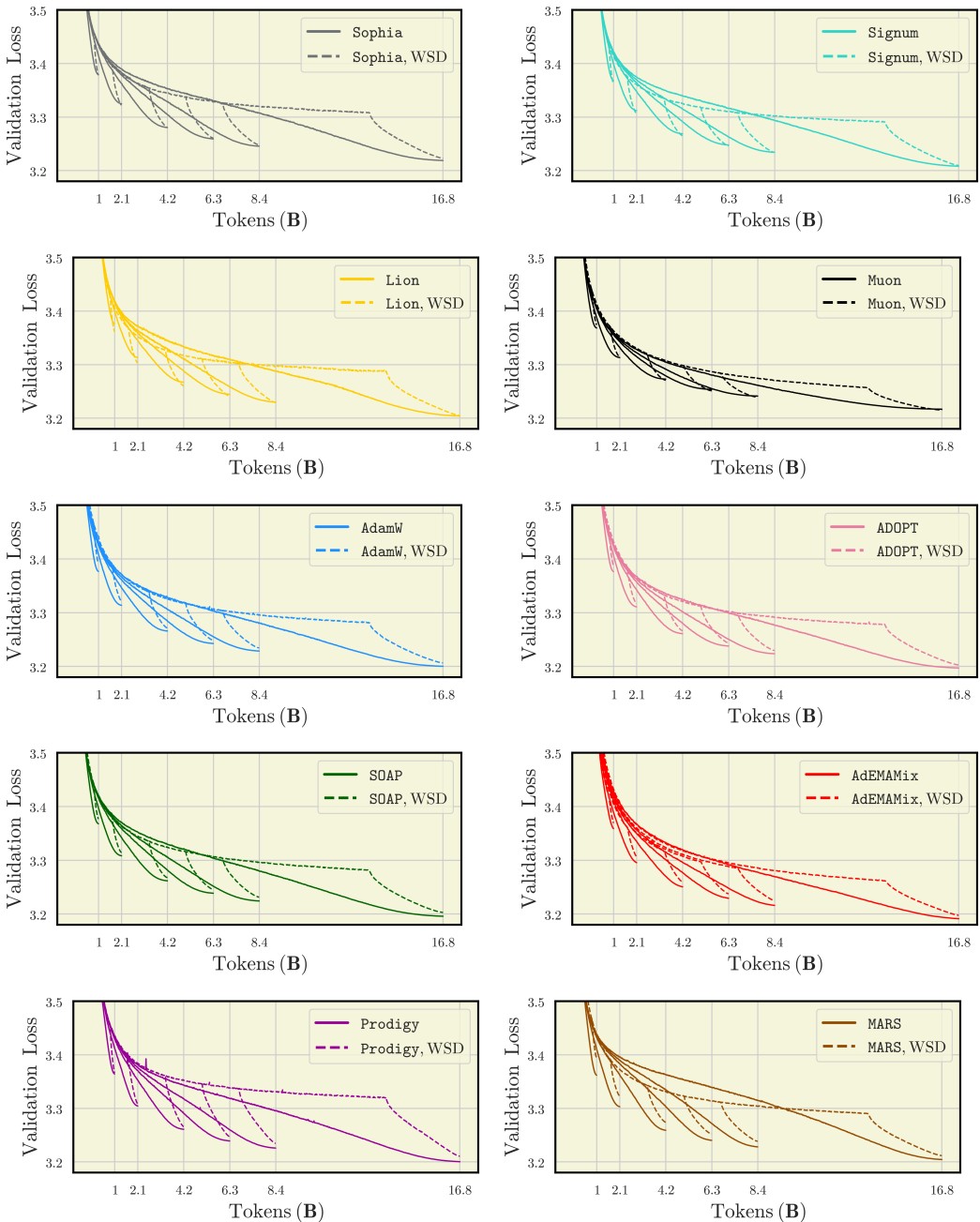

Figure 26: **WSD scheduler underperforms cosine.** We report the final validation loss on the FineWeb dataset for 124M model trained on the batch size of 256. We observe that WSD still can match the performance of cosine on `Sophia`, `Signum`, `Lion`, i.e., on the sign-based methods, and even outperform for `Muon`. Although the gap in performance is not particularly significant, but for benchmarking purposes, we decide to stick to the cosine scheduler because those gap still plays a substantial role in our setup.

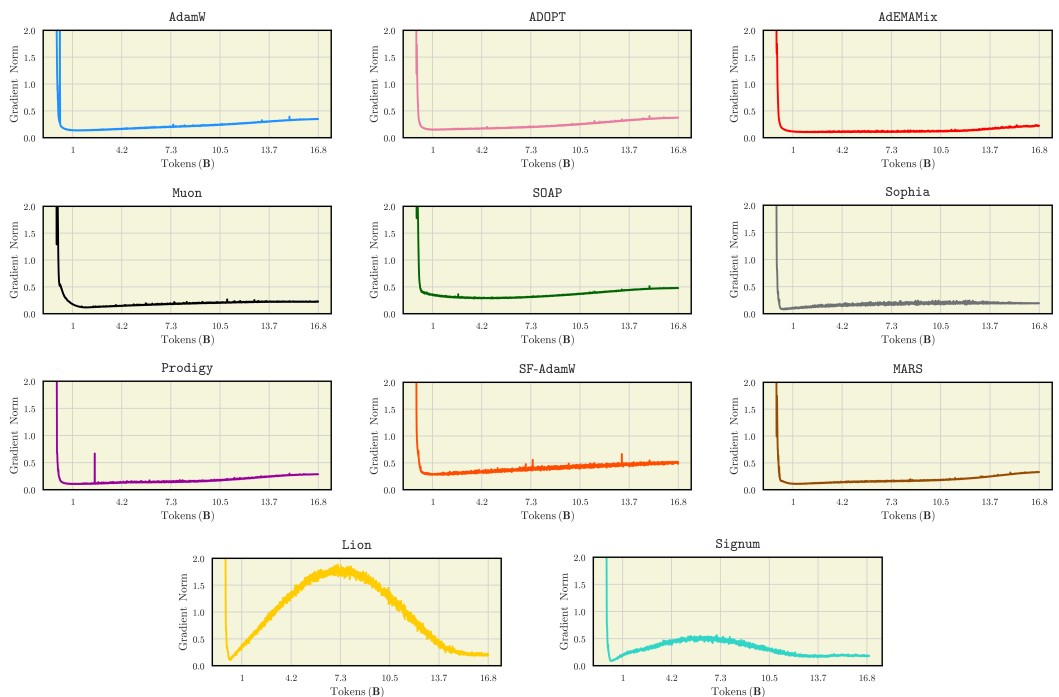

Figure 27: **Gradient Norm patterns for cosine scheduler.**

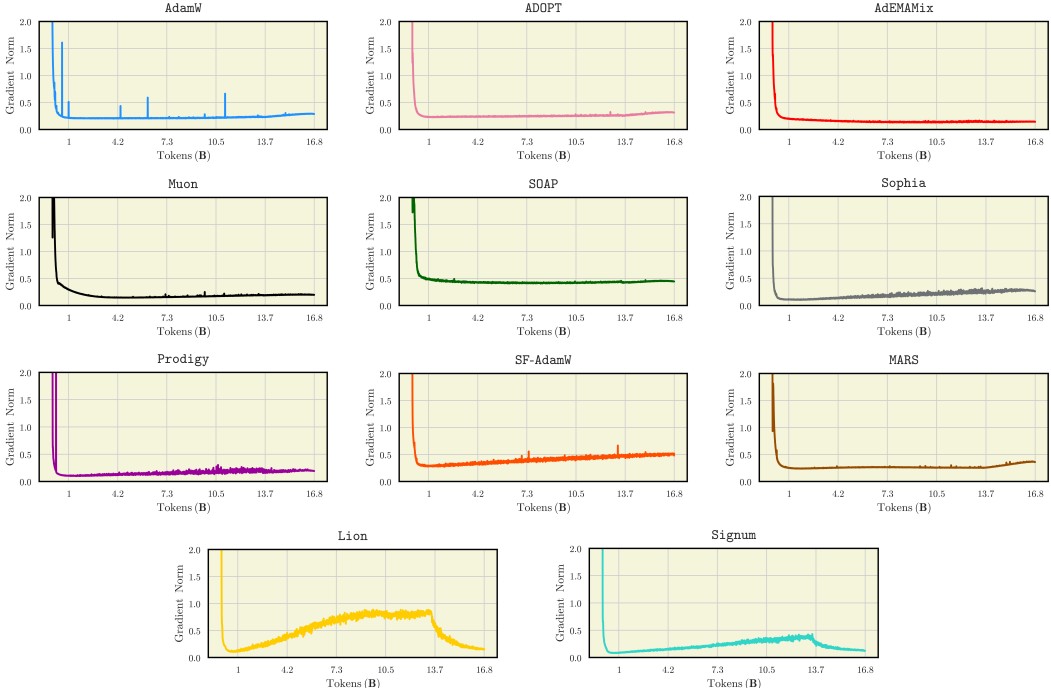

Figure 28: **Gradient Norm patterns for WSD scheduler.**

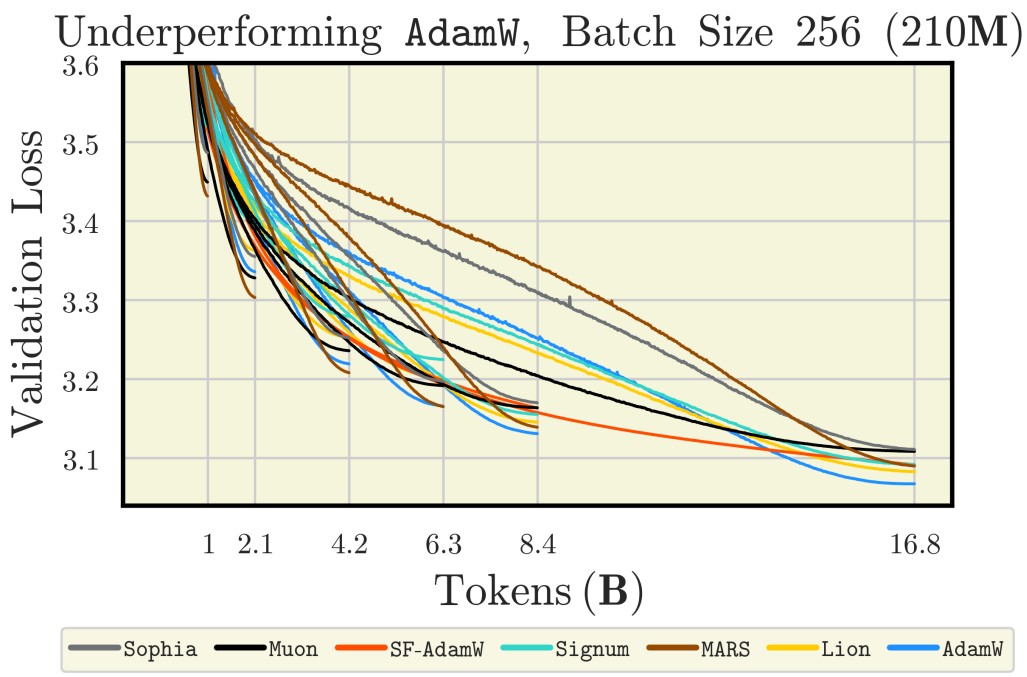

Figure 29: **Ranking of optimizers in the large-batch setting.**

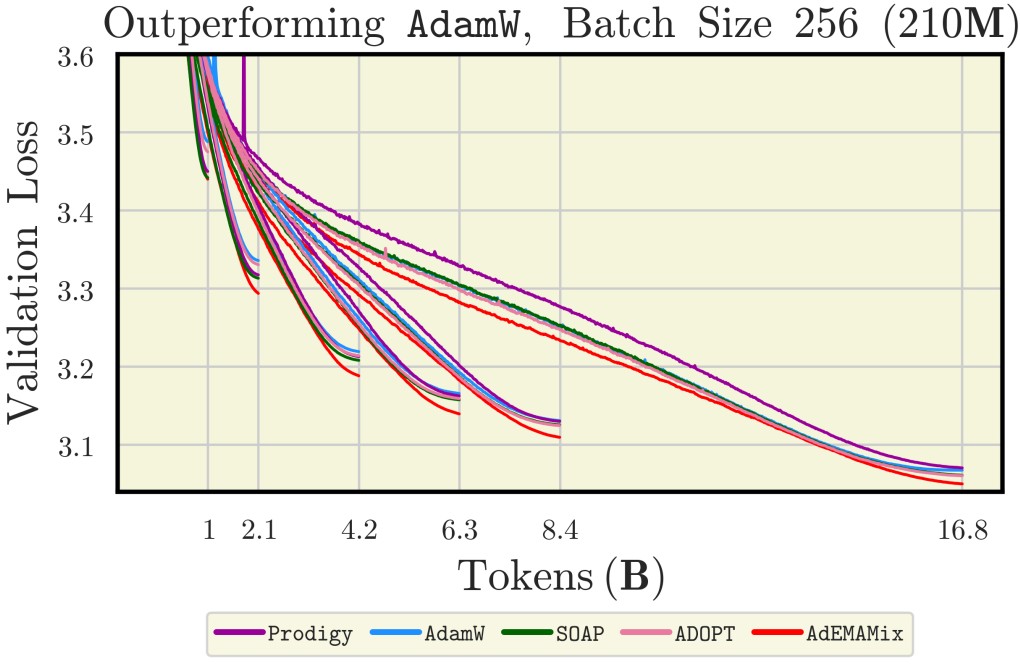

Figure 30: **Ranking of optimizers in the large-batch setting.**