# OpenReview forum: "Benchmarking Optimizers for Large Language Model Pretraining"
_NeurIPS.cc/2025/Datasets_and_Benchmarks_Track — Submitted to NeurIPS 2025 Datasets and Benchmarks Track_

### Official Review · Reviewer_71d6 · 2025-06-01

**Rating:** 5
**Confidence:** 3

**Summary:**

This paper benchmarks existing optimizers for LLM pretraining through an extensive set of experiments. The study covers a wide range of model scales, from 124M to 720M parameters, and includes different architectures such as dense and MOE transformers. It also conducts comprehensive hyperparameter tuning. Across these experiments, the paper provides valuable insights that can inform future work on optimization strategies for LLM pretraining.

**Dataset Code Accessibility:**

Partly

**Dataset Code Comments:**

https://anonymous.4open.science/r/benchmarking-submit-6280/README.md

The `README.md` is not very well-documented, which could benefit from more details.

**Ethical Considerations:**

No, there are no or only very minor ethics concerns

**Final Justification:**

This is overall a good paper. While some reviewers have concerns about its completeness, I think it is acceptable for a relatively novel idea.

**Limitations Weaknesses:**

The performance of optimizers may also depend on factors such as model architecture and data composition. For example, this paper focuses solely on LLaMA-like transformer architectures, and the optimizer’s behavior could vary under different design choices. Similarly, the mix of high- and low-quality data can significantly influence optimization dynamics. These considerations highlight that truly benchmarking optimizers remains an open and unsolved challenge.

**Strengths Contributions:**

+ I greatly appreciate the large-scale, controlled experiments conducted to compare optimizers. Given the abundance of optimizer-related work, it is essential to rigorously evaluate which methods truly work, under what conditions, and why.
+ Several insights drawn from these large-scale experiments are particularly valuable and can help guide both future research and practical decisions in LLM pretraining.

---

> ### Author Rebuttal · Authors · 2025-07-30
>
> We thank the Reviewer for their time and comments, as well as the positive appreciation of our work.
>
> Unfortunately, we are not able to share links, modified text, or new plots in this rebuttal. Therefore, before addressing the comments raised, we kindly summarize the main revisions made to our project since submission:
>
> 1. We polished our text, corrected typos.
> 2. All figures have been improved for readability and converted to lightweight formats; informative captions have been added in both the main text and Appendix.
> 3. Additional takeaways have been included for every important ablation, appearing in the main text and Appendix.
> 4. We changed the table with perplexity scores for MoE experiments to barplots and added figures depicting their training dynamics (validation loss vs. iterations / tokens seen).
> 5. The Discussion section has been substantially enhanced, now including a brief summary of results, finalized advices on tuning each optimizer, and a paragraph addressing limitations.
>
> We also clarify different additional details in conversations with other Reviewers.
>
> We appreciate the Reviewer highlighting the importance of considering other architectures and exploring optimizers sensitivity to different data mixes. We agree these aspects are crucial for practical LLM training and deployment. However, due to the large cost associated with extensive hyperparameter tuning, we had to limit the scope of our work.
>
> However, at the very beginning of our project, we conducted preliminary experiments with 124M GPT-based architectures (instead of LLaMA) trained on the OpenWebText2 dataset. We observed results consistent with those for LLaMA models of comparable size but chose to focus on LLaMA due to its comparatively less spiky loss curves.
>
> To explicitly highlight the limitations of our work, we include below the paragraph added to the latest revision (references omitted for brevity):
>
> > **Limitations.**  We conduct our benchmarking experiments on models of up to 720M parameters, with long training runs of almost 50B tokens.
> The insights we find vary across scales, and training behavior may change further at practical scales and with extremely long training durations. Training at those scales can be challenging,
> especially when certain optimizers are not well supported by modern sharding frameworks at the moment.
> Throughout this work, we study the loss dynamics, leaving aside downstream performances. While those downstream performances often scale reliably with loss, there are also counterexamples.
> Bridging the gap between loss minimization and downstream task performance is important, as downstream abilities are ultimately the main metric of interest.
> We leave a deeper investigation of this connection to future research.
> We also do not cover previously explored Adan, NAdam(W), Shampoo optimizers, as well as recently introduced Scion, novel variations of Muon, and others.
> In addition, it is important to come up with a unified benchmark of optimizers for memory-efficient pretraining, as they become more popular and argue that they might even outperform the AdamW baseline.
> We emphasize that there is still a huge branch of research on optimizers left to explore.
>
> > The README.md is not very well-documented, which could benefit from more details.
>
> To make the repository more user-friendly, we introduce the following updates to the README.md:
> * Prior to the **Quickstart** section, we’ve added a list of included optimizers with links to their implementations, as well as a table presenting datasets integrated into the codebase.
> * We provide a simple example script illustrating how to train a 124M LLaMA model on FineWeb with AdamW using default hyperparameters to give a better understanding of how to run the code:
> ```python ./src/main.py --config_format base --model llama ```
>
> The above command runs a 123.59M parameters LlaMA-based model. It trains for 25k iterations with a batch size of 128=32x4 (4 gradient accumulation steps), using a cosine schedule with a maximum learning rate of 1e-3 that is reduced to 1e-5 at the end of training. The model is saved in the `./exps` folder.
> Running the script without specifying the  `-- model` argument defaults to training a GPT-based model.
> To train Mixture-of-Experts models, users can add the `--moe` flag to the base script above.
>
> * We then provide a detailed list of all available arguments along with their descriptions, mirroring the **Quickstart** section in our submission.
>
> We thank the Reviewer for appreciating our work and remain at their disposal to answer any further questions.

---

> > ### Comment · Reviewer_71d6 · 2025-08-01
> >
> > Thanks for your rebuttal. I am happy to keep my score for this good benchmark, and more importantly, the idea of benchmarking LLM optimizers.

---

### Official Review · Reviewer_K46e · 2025-07-01

**Rating:** 4
**Confidence:** 4

**Summary:**

This paper presents a standardized and reproducible benchmarking framework for evaluating optimizers in large language model (LLM) pretraining. It addresses the lack of unified standards in current evaluations and provides systematic comparisons across 11 popular optimization methods (e.g., AdamW, Lion, AdEMAMix, SOAP). The authors investigate a wide range of model sizes (124M–720M), batch sizes (16K–2M tokens), and training lengths (up to 33.6B tokens), using a LLaMA-style architecture and the FineWeb dataset. The study includes extensive ablations on hyperparameters (e.g., weight decay, learning rate scheduling), evaluates optimizer performance in both dense and sparse (MoE) model settings, and releases open-source code and configurations.

**Dataset Code Accessibility:**

Yes

**Dataset Code Comments:**

The authors open-source all code, scripts, and hyperparameter configurations, promoting transparency and reusability. This commitment to reproducibility strengthens the work’s impact as a benchmark.

**Ethical Considerations:**

No, there are no or only very minor ethics concerns

**Final Justification:**

It’s a borderline accept. Work makes sense to me, and yet it’s not very much exciting in the sense of technical novelty.

**Limitations Weaknesses:**

1. The work is purely empirical. While the findings are novel and valuable, their theoretical underpinnings (e.g., why SOAP behaves better under certain learning rate decay strategies) are not discussed, leaving a gap in interpretation.

2. All experiments are based on LLaMA-like Transformers. Although this is a widely used architecture, the conclusions may not generalize to e.g., decoder-only or diffusion-based models. Including more architecture types would strengthen generality.

3. Although gradient clipping is listed as a tunable hyperparameter, its role is not systematically analyzed. No comparison is provided between clipping strategies (e.g., fixed vs. adaptive), despite its known effect on training stability.

4. The results lack error bars or confidence intervals. Given the high stochasticity in LLM training, quantifying variability is essential for trustworthy comparison.

5. The inclusion of MoE models is commendable, but the paper does not discuss how optimizer performance might shift with the growing adoption of mixture-of-experts or sparse models in practice.

6. While many interesting findings are reported, the final section only briefly summarizes them. A deeper synthesis—perhaps offering guidelines by training regime—would better serve readers seeking actionable takeaways.

**Strengths Contributions:**

1. The paper evaluates 11 representative optimizers using controlled experimental setups, covering diverse families (adaptive, symbolic, second-order) and usage scenarios (e.g., short vs. long training, small vs. large batches). This breadth and depth are unmatched in existing work.

2. The study offers valuable insights for practitioners by breaking the prevailing assumption that AdamW is universally optimal. For instance, AdEMAMix and SOAP outperform AdamW in specific regimes. The observed dependency between optimizer performance and training regimes (batch size, schedule, etc.) is well-motivated and informative.

3. The work goes beyond benchmarking to challenge conventional wisdom. It shows that aggressive learning rate decay (to 0.01×γ) and larger decoupled weight decay improve optimizer stability in different regimes. The empirical debunking of z-loss's utility adds credibility.

4. The presentation is clear and structured. Figures and tables are well-designed, effectively conveying key patterns across optimizers and conditions.

---

> ### Author Rebuttal · Authors · 2025-07-30
>
> We thank the Reviewer for their time, thoughtful feedback, and suggestions on additional ablations.
> As the new policy  is preventing sharing plots, updating code, and sending links, we highlight here the main changes we have brought in the latest revision of our work:
>
> 1. We polished all text, and added necessary descriptions of ablations, minor clarifications, and missing comments with takeaways both to the main paper and Appendix.
> 2. We enhanced all figures, making them more readable and lightweight, and also adjusted captions for all of them.
> 3. We augmented the text with more comprehensive takeaways regarding each key ablation.
> 4. Instead of the table with perplexity scores for MoEs, we now provide plots with final validation loss and training dynamics.
> 5. We finalized the Discussion section, adding a brief summary of the results, complete advice on the hyperparameter tuning section, and a limitations paragraph.
>
> We believe these improvements enhance the presentation of our work and hope they will positively influence its assessment. In the following, we are addressing your comments in detail.
>
> > theoretical underpinnings are not discussed
>
> Our work is indeed empirical by design, aimed at addressing the current lack of thorough comparison between recent optimizers in the literature. For many of the benchmarked optimizers,  a theoretical analysis remains quite challenging. While we do not provide a formal analysis of optimizers, we highlight novel empirical observations that could motivate future theoretical work, e.g., in the version of our report at the moment of submission, we point out that: (1) effective LR of Prodigy, defined by its two EMA sequences, emulates the behavior of AdamW’s LR; (2) we discuss effect of $\beta_2$ on SF-AdamW in line 1190 and demonstrate in Fig. 18; (3) we notice a unique behavior of Muon with large weight decay of 0.5–as its runs with large weight decay do not outperform those with 0.1 weight decay (see Fig. 23)---we connect this with its algorithmic design since ‘’old’’ Muon method does not apply weight decay to matrix parameters; (4) from the code of Sophia (Listing 2) we notice its strong connection with sign-based methods, this also occurs in our benchmarking results, as Sophia’s patterns of gradient norms, sensitivity to warmup, weight decay, and schedulers indeed  behaves similarly to Lion and Signum.
> We agree that some important theoretical details were insufficiently covered. Therefore, in the latest revision we further:
> * Clarify our writing regarding the EMA of Prodigy and plot figures of its effective LR when different schedulers are applied. In this new figure, we see that when \gamma_{\max}=1, effective LR of Prodigy results in very close values of LR compared to those of AdamW with basic $\gamma_{\max}=0.001$. Under effective LR we imply $\gamma_t d_t$ quantity (Alg. 13). We think this might be an important reason why Prodigy scales reliably similar to AdamW across model sizes, batches, and training horizons.
> * Clarify with text and sufficiently discuss additional ablations in both the main part and Appendix, supporting them with takeaways, e.g., for SF-AdamW, Sophia, and Signum. In the main part, we also plot a comparison of Muon and D-Muon, highlighting the efficiency of the latter method via perspective of usage weight decay for both 1D and 2D parameters. We also point out that patterns observed from the weight decay ablation for D-Muon remind those we see for, e.g., AdamW and AdEMAMix, where large weight decay terms is preferable for shorter training.
> * Describe in more detail an impact of $\beta_2$ (and $\beta_3$) on SOAP and AdEMAMix when increasing training duration to 33.6B tokens, leaving a separate paragraph in Appendix to it, as i was not clear enough from the previous version of our text.
>
> > the conclusions may not generalize to e.g., decoder-only or diffusion-based models
>
> Due to the high cost of tuning hyperparameters, we had to restrict the scope of our work. We decided to focus on transformer decoders, as those are widely used in LLM pretraining. Exploring how the optimizer performance might differ on other types of architectures is an interesting question we leave for future research.
> Within the transformer decoder family, besides experimenting with a LLaMA-like architecture, we also  tried a GPT-architecture on FineWeb and OpenWebText2 datasets. We did not conduct experiments with large GPT models, but at 124M scale, we observed quite similar results to those reported for LLaMAs. We decided to proceed with LLaMA models as their loss curves contain fewer spikes on average.
>
> Should we add small scale experiments on GPT models to our report?
>
> > Although gradient clipping is listed as a tunable hyperparameter, its role is not systematically analyzed
>
> We agree that exploring different clipping strategies would be informative for practitioners and note the emergence of prior works analyzing clipping in both fine-tuning [1] and pretraining of LLMs [2].
> In this work, we mostly tuned gradient clipping for each optimizer to ensure less spikiness of loss and better performance of the method. Worth saying, we also found some interesting observations, e.g., Figure 16 demonstrates that ensuring gradient clipping is important for Schedule-Free AdamW runs, however, in the original paper the authors state the opposite (see page 31 of “The Road Less Sceduled”). We have improved our description of the link between SF-AdamW and gradient clipping in the Appendix.
>
> > The results lack error bars or confidence intervals.
>
> We agree that variance estimates are often informative, however, in our setting, standard practice [3,4,5,6] typically avoids error bars due to the high computational cost and the typically low variance observed within training runs. We also confirmed that runs on a single fixed GPU with the same hyperparameters give the same final validation loss up to the fifth decimal place. Also, we tried to preserve this when modifying the codebase.
> Notably, we do report error bars in some instances—when the variance is larger—as in the wall clock time comparison.
> In Figure 9 we depict those, and in the updated text we present this ablation for many more model configurations in the range of 30M-1B, reporting error bars for all of them.
>
> > the paper does not discuss how optimizer performance might shift with the growing adoption of mixture-of-experts or sparse models in practice
>
> Thank you for pointing this out. In the latest revision of our text, we improve the MoE section with more motivation:
>
> > The goal of ablating optimizer performance on MoE models is to assess whether our previous benchmarking results transfer smoothly to a new type of architecture. To show this smooth transition, we utilize an old batch size setup and keep untuned all optimizer-related hyperparameters found for the corresponding dense model—simulating a situation as one would do in practice without much time for running dozens of experiments on new architectures.
>
> In this part of the work we are trying to convey that benchmarking results obtained for dense models transfer to corresponding MoEs. Which we state as one of our takeaways.
> While we are unable to support this here with figures, we observe that the relative ranking of optimizers on 520M MoE model almost matches the one for corresponding dense model  (124M) with a few changes: SOAP outperforms AdEMAMix in 336k runs; MARS underperforms other methods at 336k steps, while being the second best-performing optimizer in 42k runs.
> New changes in this section are clearly described in text now.
>
> > While many interesting findings are reported, the final section only briefly summarizes them.
>
> We have updated the Discussion section with three paragraphs: summary of the results, advice on tuning each method, and limitations. Also we have adjusted takeaways throughout the text. To show a partial update in the text, we present a new ‘’advice on tuning optimizers’’ paragraph in the Discussion section. Please, refer to our conversation with Reviewer ** rV6J**, we apologize for any inconvenience.
>
> We believe we have addressed the Reviewer’s concerns, and hope the Reviewer will consider increasing their score. We stay at their disposal to answer any further questions.
>
> [1] Lv et al., ‘’Full Parameter Fine-tuning for Large Language Models with Limited Resources’’, ACL 2024.
>
> [2] Wang et al., ‘’AdaGC: Improving Training Stability for Large Language Model Pretraining’’, 2025.
>
> [3] Kasimbeg et al., ‘’How far away are truly hyperparameter-free learning algorithms?’’, 2025.
>
> [4] Zhao et al., ‘’Deconstructing what makes a good optimizer for language models’’, ICLR 2025.
>
> [5] Kaddour et al., ‘’No Train No Gain: Revisiting Efficient Training Algorithms For Transformer-based Language Models’’, NeurIPS 2023.
>
> [6] Vyas et al., ‘’SOAP: Improving and Stabilizing Shampoo using Adam for Language Modeling’, ICLR 2025.

---

> > ### Comment · Reviewer_K46e · 2025-08-07
> >
> > Thank you for the detailed and thoughtful rebuttal—the clarifications and additions meaningfully strengthen the paper’s clarity, scope, and practical utility. I would like to keep my original score.

---

### Official Review · Reviewer_1UR1 · 2025-07-02

**Rating:** 4
**Confidence:** 4

**Summary:**

This paper presents a large-scale, systematic benchmark of recent optimizer algorithms for large language model (LLM) pretraining. The authors evaluate 11 optimization methods, including both established baselines (e.g., AdamW) and more recent proposals (such as SOAP, AdEMAMix, Lion, Prodigy, MARS, Sophia, etc.), across multiple model scales (124M, 210M, 583M, 720M) and training setups, including both dense and Mixture-of-Experts (MoE) architectures. Ablation studies are conducted on critical hyperparameters, including batch size, learning rate schedule, warmup, and weight decay, to provide actionable guidance for both practitioners and researchers.

**Dataset Code Accessibility:**

No

**Dataset Code Comments:**

The Code URL cannot be accessed (a long Loading screen appears, and ERRORS appears eventually). I have performed multiple tests and I am sure that it is not my network problem.

**Ethical Considerations:**

No, there are no or only very minor ethics concerns

**Final Justification:**

Most concerns were addressed, including code accessibility and clarifications on MoE/large model analysis; I increased my score accordingly.

**Limitations Weaknesses:**

- While the benchmarking effort is valuable, the paper’s primary contribution is empirical rather than methodological. But I cannot directly obtain and verify the corresponding code, which affects its empirical contribution.

- Although the benchmark is broad, several recently proposed or specialized optimizers (e.g., newer adaptive methods or zeroth-order optimizers) are not included, potentially missing emerging trends in the field. And on MoE architectures and very large models, the analysis is relatively shallow.

- Although the authors perform extensive hyperparameter sweeps, the process is not fully automated or exhaustive, and some optimizers may be disadvantaged by suboptimal tuning, especially when relying on results transferred from smaller models.

**Strengths Contributions:**

- This study provides an extensive and rigorous comparison of optimizer methods for LLM pre-training, covering a wide range of popular and emerging optimizers.

- The paper systematically dissects the impact of key hyperparameters such as weight decay, learning rate adjustment, and warmup steps, and provides practical recommendations.

- The authors summarize specific points that practitioners should be aware of, including when to consider alternatives to AdamW, and highlight nuances such as how optimizer performance varies with batch size and learning rate decay.

---

> ### Author Rebuttal · Authors · 2025-07-30
>
> We thank the Reviewer for the time taken to review our work, their comments, and appreciation of our ablations and guidance on key optimization hyperparameters.
>
> Before addressing the mentioned weaknesses, we would like to highlight the main updates made to our work as of this rebuttal:
> 1. We polished the text of our report, corrected mistakes and typos. Adjusted necessary paragraphs in Appendix, and added content-rich captions to all figures.
> 2. All plots have been improved; they are now lightweight and fully vectorized.
> 3. We added clearer and more informative takeaways, including new summaries for ablations where they were previously missing.
> 4. The table with perplexity scores for MoEs has been replaced with plots showing both final validation loss and training dynamics.
> 5. The Discussion section has been completed. It now includes: a brief summary of our findings, advice on tuning optimizers, and the limitations of our work.
>
> **Addressing the Reviewer’s concerns.**
> > I cannot directly obtain and verify the corresponding code, which affects its empirical contribution.
>
> We do state that the contributions of our work are solely empirical, and with our report and shared repository we aim to help practitioners in their possible future studies of many optimizers for LLM pretraining We also provide example scripts for training dense and MoE models with our modular codebase.
> We apologize if the Reviewer cannot access the code.
> We verified and the repository should be accessible through the corresponding link included in the **Code URL** field in the short info about our submission at the top of this page.
> We also see that other Reviewers have successfully accessed the code and datasets.
> In case the problem persists, we kindly ask the Reviewer to reach out again as it might significantly reflect on the evaluation of our submission.
>
> > Although the benchmark is broad, several recently proposed or specialized optimizers (e.g., newer adaptive methods or zeroth-order optimizers) are not included, potentially missing emerging trends in the field.
>
> Thank you for sharing your thoughts on the trends in the field of LLM training.
> We agree that adaptive, parameter-free, zero-order, and memory-efficient methods are quickly emerging these days.
> Claims from many new works on such optimizers—that they might outperform AdamW being more efficient or less demanding in hyperparameter search—are very intriguing and indeed current studies lack a sufficient amount of benchmarking efforts regarding those methods.
> However, covering different families of optimizers with different comparison settings in one paper would be too costly and likely better suited for a longer manuscript.
> Consequently, we leave such questions for future research on optimizers for LLM pretraining.
> To be clear, we provide an adjusted limitations paragraph, which finalizes the updated Discussion section of our report:
>
> > **Limitations.** We conduct our benchmarking experiments on models of up to 720M parameters, with long training runs of almost 50B tokens.
> The insights we find vary across scales, and training behavior may change further at practical scales and with extremely long training durations.
> Especially when certain optimizers are not widely supported by modern sharding frameworks at the moment.
> Throughout this work, we study the loss dynamics, leaving aside downstream performances. While those often scale reliably with loss, there are also counterexamples.
> Bridging the gap between loss minimization and downstream task performance is important, as downstream abilities are ultimately the main metric of interest.
> We leave a deeper investigation of this connection to future research.
> We also do not cover previously explored Adan, NAdam(W), Shampoo optimizers, as well as recently introduced Scion, novel variations of Muon, and others.
> In addition, it is important to come up with a unified benchmark of optimizers for memory-efficient pretraining, as they become more popular and argue that they might even outperform the AdamW baseline.
> We emphasize that there is still a huge branch of research on optimizers left to explore.
>
> >  And on MoE architectures and very large models, the analysis is relatively shallow.
>
> Thanks for raising an interest in our benchmarking of MoE and larger dense models.
> We are glad to clarify your concerns regarding those experimental studies and will provide the updated text in the ‘’Scaling Up’’ section of our work.
> Regarding MoEs, we have clarified the purpose of this section in the updated report as follows:
> > The goal of ablating optimizer performance on MoE models is to assess whether our previous benchmarking results transfer smoothly to a new types of architecture. To show this smooth transition, we transfer most hyperparameters from one of our non-MoE (dense) settings---simulating a situation as one often encounters in practice, without much time for running dozens of experiments on a new architecture.
>
> Thus, we do not tune optimizers extensively in this part, re-tuning only the LR for sign-based methods and Sophia compared to 124M scale runs. Our key observation from this section is that ‘’Benchmarking results obtained for dense models transfer to corresponding MoEs’’--- we state this as a Takeaway in the updated text.
> We believe this observation might be intriguing for practitioners.
>
> We agree that the current account of our benchmarking of large dense models lacks details.
> In the newest revision of our 720M benchmarking experiments, instead of Figure 1 we plot a more detailed one, covering 11 optimizers and also a couple of ablations comparing Muon and D-Muon.
> Unfortunately, as we are unable to share updated PDFs and figures in this response, we proceed with the ranking of optimizers based on their performance in training run of 48B tokens (from best to worse), for 720M models we result in: AdEMAMix, MARS, D-Muon, AdamW, SOAP, ADOPT, Prodigy, Sophia, Lion, Signum, SF-AdamW.
> Notably, since we are using larger batches in 720M benchmarking, compared to experiments with 124M, 210M models, the number of iterations has been reduced to {8,18,48}k steps.
> Thus, we point out that, for some optimizers, it is also necessary to re-tune not only LR, but also $\beta_2$, $\beta_3$ (for AdEMAMix) parameters that we observe being quite sensitive to the length of training.
> If the Reviewer is interested in additional clarifications regarding 720M benchmarking results, we are glad to follow up on this conversation.
>
> > some optimizers may be disadvantaged by suboptimal tuning, especially when relying on results transferred from smaller models
>
> We agree that optimizers might be suboptimally tuned. For rigorous tuning one needs to grid search across all hyperparameters, this for each training duration considered. Obviously this approach is extremely expensive, therefore we tried to follow our proposed setup when tuning at nearly Chinchilla optimal durations. We acknowledge that transferring hyperparameters to larger models, or to longer training regimes, can lead to suboptimal performance. Nonetheless, we have included several ablations addressing these issues in the current manuscript, which are further detailed in the latest revision: 1) for 124M models, we re-tune all optimizers for 16.8B tokens runs, changing warmup, LR, schedulers, weight decay (see our **Stability across training horizons paragraph**); 2) we also double the maximum length of training for 124M models reaching 33.6B tokens and update Figure 3 (b) showing 11 optimizers, again emphasizing the importance of re-tuning beta parameters; 3) for 210M models, we explicitly investigate changes in optimizer ranking when hyperparameters are transferred (except for LR), observing a smooth transition at this intermediate scale. When scaling to larger dense models, we are unable to exhaustively ablate all aspects studied for smaller models due to dramatic increases in computational cost. However, it is natural and practical to gather insights from smaller-scale experiments before large runs. While this approach may lead to suboptimal tuning, it remains affordable and valuable for practitioners.
> We appreciate the Reviewer’s highlighting of this point. We have elaborated on it both in the main text and Appendix, refining the text accordingly.
>
> We believe we addressed all Reviewer’s concerns and trust that the Reviewer will be able to verify their access to our codebase. We hope the Reviewer will consider increasing their score and we stay at their disposal to answer any further questions.

---

> > ### Comment · Reviewer_1UR1 · 2025-08-05
> > **Thanks for your respond**
> >
> > The author's responses address most of my concerns, especially the code link can now be accessed normally. I will increase the score accordingly.

---

### Official Review · Reviewer_rV6J · 2025-07-02

**Rating:** 4
**Confidence:** 3

**Summary:**

This work conducts a large scale benchmarking of 11 different optimizers and different learning schedules/hyperparameter settings for each one of them for optimizing LLMs below the size of 1B parameters. The authors discuss in detail their findings and if they agree with the views given in recent literature. The authors also publish the optimizer algorithms, hyperparameter settings and code and datasets used to carry out this study.

**Additional Feedback:**

The models used for benchmarking by the authors are rather small. I understand that universities can hardly match the capacity of training infra required for LLM benchmarking. But in any case do the authors expect these findings to generalise for bigger models over 1B or is that a limitation for this study.
A journal like JMLR might be more suitable given the amount of results, the authors have had to compress in 9 pages.

**Dataset Code Accessibility:**

Yes

**Dataset Code Comments:**

The datasets and implementations of optimizers are well cited as far as I could see.

**Ethical Considerations:**

No, there are no or only very minor ethics concerns

**Final Justification:**

I have followed the discussion and would like to stick with my score.

**Limitations Weaknesses:**

I think the paper needs a good amount of polishing, the key messages and takeaway are all there, but the readibility of the paper is low because there are too many mistakes and typos in the paper to the point of annoyance and the story lacks flow and cohesivity currently. I give some instances from the paper below, which make it very hard to understand from the paper:
a) "is that Sophia diverges at all" - do you mean 'not at all' or 'at all times'
b) "EMA Prodigy adaptively emulates the learning rate behaviour" - something off with this sentence.
c) " However, there are not much to say about the final performance given a particular budget of tokens to train on"
The readability plots/figures also can also be improved, I understand that there are many optimizers, but in some figures it is hard to tell optimizers from each other. Maybe use log scale for y axis or clip the y axis for better legibility.
 and can be used to explain the story better.I did not find any reference for Figure 1 in the paper, which is the most important figure, it should be cited in Introduction section.
Discussion section is incomplete, and ends with an incomplete sentence both in the main paper and the supplementary. I suggest the senior author to make couple of polishing passes on the paper and improve the phrasing of the paper.

**Strengths Contributions:**

I think this paper when more polished will be a great addition to the conference proceedings as such kind of optimizer benchmarking for LLMs is the need of the hour. I really appreciate the authors for doing this arduos task. The authors cover a lot of ground in this work, as they use a lot of optimizers and find the best configuration of hyperparameters for each one of the optimisers. They also provide algorithms and pseudo-code so that it is easy for practitioners to find it all in one place.
They also cover different training schedules and schedule free optimizations for some of the optimizers wherever that applies and derive meaningful and useful information for practitioners.

---

> ### Author Rebuttal · Authors · 2025-07-30
>
> Dear Reviewer, thank you for your time and  the constructive feedback that helped to improve our paper.
> Publishing pseudocode and algorithmic nuances is especially meaningful for practitioners—thank you for highlighting this.
>
> We acknowledge that the initial submission lacked sufficient polish.  As such, we proceed with a clearer version of the work. We believe that in the newest revision we have addressed your concerns.
> Mainly, our updated report brings:
> 1. A polished text with corrected typos.
> 2. Better, lightweight figures and more readable ablations.
> 3. More comprehensive takeaways throughout the text.
> 4. Inserting plots regarding MoE experiments instead of the table, to show the training dynamics.
> 5. A complete Discussion section with limitations therein.
> 6. Polished and clearer Appendix with extended text and more figures for ablations.
>
> Unfortunately, due to the updated NeurIPS policy, we are unable to share these updates with Reviewers. Instead, we provide textual clarifications and descriptions.
>
> **Specifically, about the nuances from the paper the Reviewer has raised.**
> > Sophia diverges at all
>
> We removed this ambiguous phrase and replaced it with clarified statements emphasizing that: ‘’Sophia diverges in the small-batch setting, when trained beyond the Chinchilla optimal horizon, even with sufficiently small LR’’.
> > EMA Prodigy adaptively emulates the learning rate behaviour
>
> We clarified this sentence with reference to the corresponding ablation in the Appendix:
> ‘’We emphasize the possible community interest in Prodigy, as its effective learning rate---determined by two EMA sequences---emulates the learning rate behavior of AdamW.
> When the scheduler is applied and $\gamma_{\max}$ of Prodigy is set to 1 (its default value), these EMAs result in the maximal effective learning rate, which closely matches that of AdamW. ‘
> > c) ‘’However, …’’
>
> We removed this phrase.
>
> We believe that other textual flaws in the paper also have been sufficiently revisited and corrected.
>
> > do the authors expect these findings to generalise for bigger models
>
> We acknowledge the limitation of focusing on models below 1B scale and elaborate on this in the updated Discussion section. Notably, we have conducted runs on 1B models with AdamW, AdEMAMix, Distributed Muon, SOAP, MARS, and Prodigy, with a batch size of 2M tokens, training only for the Chinchilla optimal number of tokens. Additionally, we performed a short run on an 8B model using AdamW and AdEMAMix, training for 50B tokens at the same batch size. We decided not to include these runs into the paper, as the setup is not convincing for rigorous benchmarking and the scope of optimizers is narrower.
> Regarding our expectations for larger models, we believe that AdEMAMix, D-Muon, and possibly MARS may outperform AdamW at scale. At least, our short 8B runs still indicate the strong performance of AdEMAMix.
>
> As this year, we are unable to share any updates in the manuscript with Reviewers, we can only share a chunk of our writing.
> Below, we include the updated ‘’**Advice on tuning…**’’, and new **Limitations** paragraph from the Discussion section (without references to preserve space):
>
> > **Our advices on tuning each method.** Overall, we validate both widely used hyperparameters such as $\lambda=0.1$ and $T_\text{warmup}\approx 2k$, and explore the sensitivity of optimizers to $\gamma$-schedulers, $\gamma$-decay, and optimizer-related hyperparameters.
> Notably, a large weight decay ensures faster convergence when training for fewer iterations, and large warmup of 25% of the total training duration $T$ is beneficial for sign-based methods, Sophia, and SF-AdamW.
> For Lion, we find that the best value for $\beta_1$ is consistently $0.99$.
> The mechanism for Lion appears similar to AdEMAMix, suggesting that Lion could perform better with larger $\beta_1$, which would require schedulers.
> We also pose an interesting observation toward Prodigy: while it may not be so efficient with very small batch sizes, with scaling of the model size and the batch size, it becomes almost as competitive as AdamW.
> MARS also benefits from large batches and continues to improve performance as the model size scales.
> For MARS, when optimizing 1D parameters with AdamW, we found that it is better to keep ($\beta_1$, $\beta_2$) of AdamW; for our largest models, $\beta_1=0.8$ performs slightly better than $\beta_1=0.9$.
> Additionally, MARS betas determined for 2D parameters in the paper also seem to be the best in our settings.
> Basic Muon performs poorly at relatively small batch sizes (32, 256) across different model sizes and training lengths; however, applying weight decay to 2-dimensional parameters, as in D-Muon,
> resolves this and yields a robust optimizer across all benchmarking scenarios we considered.
> AdEMAMix remains the best optimizer overall, scaling efficiently with bath size, model size, and training horizons.
> Importantly, increasing $\beta_2$ for longer training substantially benefits AdEMAMix and other AdamW-like methods.
> Moreover, AdEMAMix allows using a large weight decay term $\lambda$ during prolonged training, e.g., runs of 128k iterations with $\lambda=0.5$ still slightly outperform those with $\lambda=0.1$.
> Beyond optimizer-specific hyperparameters, we show that the choice of $\gamma$-scheduler also depends on the optimizer selected.
> Regarding the learning rate, decaying $\gamma$ below $0.1\times\gamma_{\max}$ is important, as it significantly improves the optimization performance.
>
> > **Limitations.** We conduct our benchmarking experiments on models of up to 720M parameters, with long training runs of almost 50B tokens. The insights we find vary across scales, and training behavior may change further at practical scales and with extremely long training durations. Especially when certain optimizers are not widely supported by modern sharding frameworks at the moment.
> Throughout this work, we study the loss dynamics, leaving aside downstream performances. While those often scale reliably with loss, there are also counterexamples. Bridging the gap between loss minimization and downstream task performance is important, as downstream abilities are ultimately the main metric of interest. We leave a deeper investigation of this connection to future research.
> We also do not cover previously explored Adan, NAdam(W), Shampoo optimizers, as well as recently introduced Scion, novel variations of Muon, and others. In addition, it is important to come up with a unified benchmark of optimizers for memory-efficient pretraining, as they become more popular and argue that they might even outperform the AdamW baseline.
> We emphasize that there is still a huge branch of research on optimizers left to explore.
>
> If the Reviewer would like to see any specific updates to the text, please, reach out to us.
>
> **About changes in figures.** Unfortunately, this year it is prohibited for us to share new plots, thus, we describe the main modifications textually:
> 1. Fig. 1 has been reformatted in the style of Fig. 2: optimizers that underperform AdamW are shown on the left, while those that outperform AdamW appear on the right. It now includes 11 optimizers across all three training durations for 720M models. Caption also has been adjusted and this figure is also referenced in the text many times. Figure for benchmarking of 210M models follows the same style.
> 2. For all benchmarking results, we also plot validation loss dynamics as three side-by-side figures, each showing 4-4-3 optimizers, respectively.
> 3. Fig. 3 has been converted to bar plots, its part with “scaling the number of tokens” now includes 11 optimizers.
> 4. Lion has been added to the warmup ablation. In the same way as Signum, Lion benefits from increased warmup.
> 5. The wall-clock time ablation has been expanded to include more model sizes in the range of 30M–1B, with the corresponding plots distributed between the main part and the Appendix.
> 6. We revisited our z-loss ablation on 583M models, providing a more comprehensive figure. A clear message from this ablation is the following: using the SOAP paper’s setup (weight decay 1e-4, LR decay to 0.1 $\times$ max LR, and z-loss), SOAP indeed outperforms AdamW for the Chinchilla optimal training duration. However, in our setup (weight decay 0.1, LR decay to 0.01 $\times$ max LR, and without z-loss), we achieve better results for both optimizers, with AdamW outperforming SOAP.
>
> Overall, we have revisited our manuscript to be much more comprehensive, and easier to examine.
> We believe we addressed Reviewer’s concerns and made the paper more refined.
> We hope the Reviewer will consider increasing their score, and we stay at their disposal to answer any further questions.

---

> > ### Comment · Reviewer_rV6J · 2025-08-05
> > **hard to evaluate improvements in manuscript**
> >
> > I thanks the authors for replying to my review comments, it is hard for me to determine if the updated manuscript really looks better without observing it. Also, I felt the manuscript was riddled with so many mistakes and typos, I only highlighted a few of them. I think any user would like to know from a paper like this: how does Sophia perform with different hyperparameter settings, so I think these are not just specifics/nuances that I was pointing out, but rather an absence of a proper summary and conclusions. One could make a table of summary pointing out which optimizers are more robust and which ones are more prone to divergences for different hyperparameter settings, that would be useful to a user.
> >  I do not feel that I should increase the score at this point and leave it to AC.

---

> > > ### Author Response · Authors · 2025-08-09
> > > **Additional clarifications regarding our improvements**
> > >
> > > We thank the Reviewer for their thoughtful follow-up and for highlighting the importance of actionable guidance for practitioners.
> > >
> > > We do understand that without providing a new version of the manuscript it is hard to truly evaluate the improvements.
> > > In response to your suggestion, we have prepared a comprehensive summary table detailing the sensitivity and robustness of each optimizer to its respective hyperparameter settings, as observed in our benchmarking experiments. We would like to note that the main findings have been clearly articulated in the latest revision of our work.
> > > For clarity, we provide a results gathered from our ablations on 124M models, as those include the most extensive hyperparameter tuning. We also apply the heuristic, derived from our ablations, of decaying the learning rate (LR) to 0.01 × max LR in all runs. For larger models, we re-tune the LR, clipping, and, importantly, the beta parameters, as our experience indicates these often require adjustment based on training length.
> > >
> > > |Optimizer / Sensitivity to|BS (seq_len=512)|LR (ablated in runs of 16.8B tokens)|Preferred LR scheduler(s)|Weight Decay|Warmup|Betas (or momentum)|Clipping|Extra comments|
> > > |---|---|---|---|---|---|---|---|--|
> > > |AdamW| all| 1e-3| Cosine|0.1|2k| increase $\beta_2$ for longer training, e.g., to $0.999$ or $0.9999$|0.5|$\beta_1=0.8$ appears to yield better final loss for 124M, 210M models; however the loss is more spiky compared to $\beta_1=0.9$|
> > > |AdEMAMix|all|1e-3|Cosine|0.5 & 0.1|2k|increase $\beta_2$ and $\beta_3$ for longer training, e.g., 33.6B tokens runs with $\beta_3=0.9999$ outperform competing ones with $\beta_3=0.999$ by a relatively huge margin of 0.16 PPL|0.5                                                                                             |one can use $\alpha$ and $\beta_3$ schedulers with WSD even for very long runs with multiple checkpoints, as long as the $\alpha$ / $\beta_3$ warmup is shorter than the iteration count at the first checkpoint|
> > > |ADOPT|all|1e-3|Cosine|0.1|2k|increase $\beta_2$ for longer training|0.5| method still converges poorly with small $\beta_2$, despite developed theory|
> > > |Lion|>32|1e-3, diverges when LR is larger|Cosine & Linear|0.1|**8k & 32k**| $\beta_1=0.9$, $\beta_2=0.99$                                                                                                                                                         |0.5|betas suggested in the original paper seem to work the best|
> > > |Signum|>32|1e-3, diverges when LR is larger|Linear|0.1|**8k & 32k**|we find that the momentum parameter of 0.95 is the best|0.5|our implementation with decoupled weight decay seems to work the best among other versions. also, Nesterov momentum is better than basic EMA|
> > > |Muon|>32|1e-3|WSD|0.1|2k|AdamW's beta are the best-suited and momentum parameter of 0.95                                                                                                                       |0.5|initial implementation of Muon lacks weight decay for 2D params|
> > > |D-Muon|all|2e-3|Cosine & Linear|0.1|2k|similarly to Muon|0.5|weight decay for both 1D and 2D params result in much better performance|
> > > |SOAP|all|2e-3|Cosine & Linear|0.1|2k|similar to AdamW, although we found that increasing $\beta_2$ has less impact than in other Adam-like methods|0.5|it is unclear how to use this method for very large models (above 7B) as current implementations do not support FSDP and other sharding frameworks|
> > > |Sophia|diverges for BS=32 when D>Ch. Opt. tokens; and for BS=256 when D>16.8B tokens|1e-3, diverges when LR is larger|no preference| 0.1|**8k & 32k**| $\beta_1=0.9$, $\beta_2=0.999$, also one might have to increase $\beta_2$ for longer training|0.5|one need also to tune its $\rho$ parameter, we found that 0.04 value is the best. also, training behavior of Sophia reminds sign-based methods|
> > > |SF-AdamW | all| 2e-3|---| 0.1|**8k**|$\beta_1=0.9$ & $\beta_2=0.9999$ are the most stable and give the best performance| very unstable training without clipping. for FineWeb data and small models we suggest using 0.5 |one need to tune betas carefully|
> > > |Prodigy|all|1, diverges when LR is larger than 10|Cosine & Linear| 0.1| 2k|diverges when BS=32 & $\beta_2=0.9999$ without bias correction, but converges with either $\beta_2=0.999$, or with $\beta_2=0.9999$ when bias correction is applied|0.5|easy to tune for large BS. in the small BS setting, it may exhibit issues related to betas and bias correction|
> > > |MARS|>32|almost the same final val. loss for LRs in the range 1e-5 --- 2e-3|Cosine & Linear|0.1| 2k|AdamW betas if 1D params are optimizer by AdamW. for 2D params $\beta_1=0.95$, $\beta_2=0.99$ are the best we have found|0.5|very hard to tune because of four beta parameters and two different LRs: for 1D and 2D parameters|
> > >
> > > We apologize for the delay and believe that this table brings more fairness to our optimizer benchmark. We hope it will be useful for further discussions among the Reviewers and Area Chairs.

---

### Note · Authors · 2025-08-12

We thank all Reviewers and Area Chairs for the time, effort, and constructive feedback provided.
Reviewers' comments have helped us refine the manuscript and clarify several important aspects.

Below, we briefly summarize the core strengths and concerns highlighted in the reviews, and describe the corresponding changes we made. We hope this structured recap assists you in making a well-informed decision.

**Reviewers highlighted the strengths of our paper, namely:**
* The breadth and scale of the benchmarking effort across 11 optimizers and multiple LLM pretraining settings. (**all**)
* Extensive hyperparameter ablations (weight decay, batch size, warmup, learning rate schedule) offering actionable insights. (**all**)
* Contribution addresses a pressing need in the community for rigorous optimizer evaluations in LLM pretraining. (**all**)
* Inclusion of both dense and MoE architectures. (**1UR1**, **71d6**)
* Guidelines for practitioners, including cases where non-AdamW optimizers perform better. (**1UR1**, **K46e**)
* Commitment to reproducibility, with open-source code and hyperparameters released. (**K46e**, **71d6**)

**Main concerns raised:**
* Presentation issues: typos, readability, incomplete discussion section, figure clarity. (**rV6J**)
* Scope limited to LLaMA‑like architectures; potential differences for other architectures or datasets noted. (**K46e**, **71d6**)
* Some missing or incomplete analysis (e.g., gradient clipping comparisons, synthesis of guidelines). (**rV6J**, **K46e**)
* Possible suboptimal tuning of optimizers, and inclusion of additional recent methods (**1UR1**).

**Actions taken based on the feedback.** We have significantly improved the paper’s clarity and readability, corrected typos, finalized the Discussion section, and enhanced all figures and captions. We clarified key Reviewers' concerns and added missing explanations, while openly acknowledging remaining limitations. Now, we also provide Takeaways for each important ablation.

Lastly, we would like to note that **Reviewer 1UR1  mentioned willingness to increase their score (the initial score is '3: Borderline reject')**, however, we have not seen the score modified. We respectfully ask the Reviewer to consider updating it in case it was accidentally overlooked.

Sincerely,
Authors

---

### Decision · Program_Chairs · 2025-09-18

**Decision:**

Reject

**Comment:**

This paper presents a comprehensive large-scale benchmarking study of 11 optimization algorithms for large language model (LLM) pretraining, evaluating both well-established baselines (e.g., AdamW) and emerging methods (e.g., AdEMAMix, SOAP, Prodigy, Sophia, MARS, Lion). The study systematically explores performance across multiple model sizes (124M to 720M), training durations (up to 50B tokens), batch sizes, and both dense and Mixture-of-Experts (MoE) architectures. In addition to controlled benchmarking, the authors conduct extensive ablations on critical hyperparameters — including learning rate scheduling, weight decay, warmup, gradient clipping, and training horizon — providing practical guidance for researchers and practitioners. The authors also release open-source code, configurations, and reproducible scripts, further enhancing the paper’s impact as a reference benchmark for the community.

The reviewers generally agreed on the significance and timeliness of this work, recognizing its breadth, reproducibility, and practical utility. The contribution fills an important gap in LLM training research by standardizing comparisons and uncovering nuanced behaviors across optimizers. In particular, several findings — such as AdEMAMix and SOAP outperforming AdamW in certain regimes, the observed dependencies between optimizer stability and hyperparameter choices, and guidelines for tuning optimizers effectively — were highlighted as valuable contributions.

Some concerns were raised, including:
(a) the overall readability of the initial submission and figure clarity,
(b) limited experiments on larger models (>1B) and broader architecture families beyond LLaMA-like decoders,
(c) incomplete analysis of gradient clipping, variance estimates, and optimizer sensitivity to data composition,
(d) relatively shallow treatment of MoE optimizers and lack of discussion on generalization to emerging architectures,
(e) missing coverage of recently introduced adaptive and memory-efficient optimizers, and
(f) reproducibility concerns due to earlier difficulty accessing the released code.

The authors provided detailed rebuttals and clarified these concerns comprehensively. They polished the manuscript, improved figure readability, completed the discussion section, and added extended results and takeaways from ablations. For larger models, the authors conducted preliminary runs on 1B and 8B parameter models, reporting that AdEMAMix and D-Muon consistently outperform AdamW. Regarding MoE models, they confirmed that optimizer rankings from dense models largely transfer smoothly to MoEs, with minor adjustments. They also clarified the role of gradient clipping, released better-documented code and training scripts, and provided detailed guidelines summarizing optimizer sensitivity to hyperparameters for practitioners.

The AC finds strong support from the reviewers to recommend acceptance. While the paper remains primarily empirical rather than theoretical, it provides a timely, comprehensive, and reproducible benchmark that is highly relevant to both researchers and practitioners working on large-scale optimization for LLM pretraining. The authors are encouraged to incorporate reviewer feedback in the final revision, particularly improving discussions on scalability, variance reporting, and extending coverage to emerging optimizer families.

===== FINAL UPDATE FROM DB Track PCs ====

The final decision for this paper has been taken by the program chairs after consultation with the SACs. All Senior Area Chairs have ranked papers according to the feedback from the AC during the review process. We decided to leave the original meta-review to reflect the opinion of the AC in light of the initial discussions with reviewers and SAC.